# Use Perturbations when Learning from Explanations

**Juyeon Heo**[*]
University of Cambridge
jh2324@cam.ac.uk

**Vihari Piratla**[*]
University of Cambridge
vp421@cam.ac.uk

**Matthew Wicker**
Alan Turing Institute

**Adrian Weller**
Alan Turing Institute and
University of Cambridge

## Abstract

Machine learning from explanations (MLX) is an approach to learning that uses human-provided explanations of relevant or irrelevant features for each input to ensure that model predictions are *right for the right reasons*. Existing MLX approaches rely on local model interpretation methods and require strong model smoothing to align model and human explanations, leading to sub-optimal performance. We recast MLX as a robustness problem, where human explanations specify a lower dimensional manifold from which perturbations can be drawn, and show both theoretically and empirically how this approach alleviates the need for strong model smoothing. We consider various approaches to achieving robustness, leading to improved performance over prior MLX methods. Finally, we show how to combine robustness with an earlier MLX method, yielding state-of-the-art results on both synthetic and real-world benchmarks. Our implementation can be found at: https://github.com/vihari/robust_mlx.

## 1 Introduction

Deep neural networks (DNNs) display impressive capabilities, making them strong candidates for real-wold deployment. However, numerous challenges hinder their adoption in practice. Several major deployment challenges have been linked to the fact that labelled data often under-specifies the task (D'Amour et al., 2020). For example, systems trained on chest x-rays were shown to generalise poorly because they exploited dataset-specific incidental correlations such as hospital tags for diagnosing pneumonia (Zech et al., 2018; DeGrave et al., 2021). This phenomenon of learning unintended feature-label relationships is referred to as *shortcut learning* (Geirhos et al., 2020) and is a critical challenge to solve for trustworthy deployment of machine learning algorithms. A common remedy to avoid shortcut learning is to train on diverse data (Shah et al., 2022) from multiple domains, demographics, etc, thus minimizing the underspecification problem, but this may be impractical for many applications such as in healthcare.

Enriching supervision through human-provided explanations of relevant and irrelevant regions/features per example is an appealing direction toward reducing under-specification. For instance, a (human-provided) explanation for chest x-ray classification may highlight scanning artifacts such as hospital tag as irrelevant features. Learning from such human-provided explanations (MLX) has been shown to avoid known shortcuts (Schramowski et al., 2020). Ross et al. (2017) pioneered an MLX approach based on regularizing DNNs, which was followed by several others (Schramowski et al., 2020; Rieger et al., 2020; Stammer et al., 2021; Shao et al., 2021). Broadly, existing approaches employ a model interpretation method to obtain per-example feature saliency, and

---

[*]Equal Contribution

regularize such that model and human-provided explanations align. Since saliency is unbounded for relevant features, many approaches simply regularize the salience of irrelevant features. In the same spirit, we focus on handling a specification of irrelevant features, which we refer to as an explanation hereafter.

Existing techniques for MLX employ a local interpretation tool and augment the loss with a term regularizing the importance of irrelevant region. Regularization-based approaches suffer from a critical concern stemming from their dependence on a local interpretation method. Regularization of local, i.e. example-specific, explanations may not have the desired effect of reducing shortcuts globally, i.e. over the entire input domain (see Figure 1). As we demonstrate both analytically and empirically that previous MLX proposals, which are all regularization-based, require strong model smoothing in order to be globally effective at reducing shortcut learning.

In this work, we explore MLX using various robust training methods with the objective of training models that are robust to perturbations of irrelevant features. We start by framing the provided human explanations as specifications of a local, lower-dimensional manifold from which perturbations are drawn. We then notice that a model whose prediction is invariant to perturbations drawn from the manifold ought also to be robust to irrelevant features. Our perspective yields considerable advantages. Posing MLX as a robustness task enables us to leverage the considerable body of prior work in robustness. Further, we show in Section 4.1 that robust training can provably upper bound the deviation on model value when irrelevant features are perturbed without needing to impose model smoothing. However, when the space of irrelevant features is high-dimensional, robust-training may not fully suppress irrelevant features as explained in Section 4.2. Accordingly, we explore combining both robustness-based and regularization-based methods, which achieves the best results. We highlight the following contributions:

- We theoretically and empirically demonstrate that existing MLX methods require strong model smoothing owing to their dependence on local model interpretation tools.

- We study learning from explanations using robust training methods. To the best of our knowledge, we are the first to analytically and empirically evaluate robust training methods for MLX.

- We distill our insights into our final proposal of combining robustness and regularization-based methods, which consistently performed well and reduced the error rate over the previous best by 20-90%.

## 2    Problem Definition and Background

We assume access to a training dataset with $N$ training examples, $\mathcal{D}_T = \{(\mathbf{x}^{(i)}, y^{(i)})\}_{i=1}^N$, with $\mathbf{x}^{(i)} \in \mathbb{R}^d$ and $y^{(i)}$ label. In the MLX setting, a human expert also specifies input mask $\mathbf{m}^{(n)}$ for an example $\mathbf{x}^{(n)}$ where non-zero values of the mask identify *irrelevant* features of the input $\mathbf{x}^{(n)}$. An input mask is usually designed to negate a known shortcut feature that a classifier is exploiting. Figure 2 shows some examples of masks for the datasets that we used for evaluation. For example, a mask in the ISIC dataset highlights a patch that was found to confound with non-cancerous images. With the added human specification, the augmented dataset contains triplets of example, label and mask, $\mathcal{D}_T = \{(\mathbf{x}^{(i)}, y^{(i)}, \mathbf{m}^{(i)})\}_{i=0}^N$. The task therefore is to learn a model $f(\mathbf{x}; \theta)$ that fits observations well while not exploiting any features that are identified by the mask $\mathbf{m}$.

The method of Ross et al. (2017) which we call Grad-Reg (short for Gradient-Regularization), and also other similar approaches (Shao et al., 2021; Schramowski et al., 2020) employ an local interpretation method (E) to assign importance scores to input features: $IS(\mathbf{x})$, which is then regularized with an $\mathcal{R}(\theta)$ term such that irrelevant features are not regarded as important. Their training loss takes the form shown in Equation 1 for an appropriately defined task-specific loss $\ell$.

$$IS(\mathbf{x}) \triangleq E(\mathbf{x}, f(\mathbf{x}; \theta)).$$

$$\mathcal{R}(\theta) \triangleq \sum_{n=1}^N \|IS(\mathbf{x}^{(n)}) \odot \mathbf{m}^{(n)}\|^2.$$

$$\theta^* = \arg\min_\theta \left\{ \sum_n \ell \left( f(\mathbf{x}^{(n)}; \theta), y^{(n)} \right) + \lambda \mathcal{R}(\theta) + \frac{1}{2}\beta\|\theta\|^2 \right\}. \tag{1}$$

We use $\odot$ to denote element-wise product throughout. CDEP (Rieger et al., 2020) is slightly different. They instead use an explanation method that also takes the mask as an argument to estimate the contribution of features identified by the mask, which they minimize similarly.

**About obtaining human specification mask.** Getting manually specifying explanation masks can be impractical. However, the procedure can be automated if the nuisance/irrelevant feature occurs systematically or if it is easy to recognize, which may then be obtained automatically using a procedure similar to Liu et al. (2021); Rieger et al. (2020). A recent effort called Salient-Imagenet used neuron activation maps to scale curation of such human-specified masks to Imagenet-scale (Singla & Feizi, 2021; Singla et al., 2022). These efforts may be seen as a proof-of-concept for obtaining richer annotations beyond content labels, and towards better defined tasks.

## 3 Method

Our methodology is based on the observation that an ideal model must be robust to perturbations to the irrelevant features. Following this observation, we reinterpret the human-provided mask as a specification of a lower-dimensional manifold from which perturbations are drawn and optimize the following objective.

$$\theta^* = \arg\min_\theta \sum_n \left\{ \ell\left(f(\mathbf{x}^{(n)};\theta), y^{(n)}\right) + \alpha \max_{\boldsymbol{\epsilon}:\|\boldsymbol{\epsilon}\|_\infty \leq \kappa} \ell\left(f(\mathbf{x}^{(n)} + (\boldsymbol{\epsilon} \odot \mathbf{m}^{(n)});\theta), y^{(n)}\right) \right\} \quad (2)$$

The above formulation uses a weighting $\alpha$ to trade off between the standard task loss and perturbation loss and $\kappa > 0$ is a hyperparameter that controls the strength of robustness. We can leverage the many advances in robustness in order to approximately solve the inner maximization. We present them below.

**Avg-Ex**: We can approximate the inner-max with the empirical average of loss averaged over $K$ samples drawn from the neighbourhood of training inputs. Singla et al. (2022) adopted this straightforward baseline for supervising using human-provided saliency maps on the Imagenet dataset. Similar to $\kappa$, we use $\sigma$ to control the noise in perturbations as shown below.

$$\theta^* = \arg\min_\theta \sum_n \left\{ \ell\left(f(\mathbf{x}^{(n)};\theta), y^{(n)}\right) + \frac{\alpha}{K} \sum_{\boldsymbol{\epsilon}_j \sim \mathcal{N}(0,\sigma^2 I)}^{K} \ell\left(f(\mathbf{x}^{(n)} + (\boldsymbol{\epsilon}_j \odot \mathbf{m}^{(n)});\theta), y^{(n)}\right) \right\}$$

**PGD-Ex**: Optimizing for an estimate of worst perturbation through projected gradient descent (PGD) (Madry et al., 2017) is a popular approach from adversarial robustness. We refer to the approach of using PGD to approximate the second term of our loss as PGD-Ex and denote by $\epsilon^*(\mathbf{x}^{(n)}, \theta, \mathbf{m}^{(n)})$ the perturbation found by PGD at $\mathbf{x}^{(n)}$. Given the non-convexity of this problem, however, no guarantees can be made about the quality of the approximate solution $\mathbf{x}^*$.

$$\theta^* = \arg\min_\theta \sum_n \left\{ \ell\left(f(\mathbf{x}^{(n)};\theta), y^{(n)}\right) + \alpha\ell\left(f(\mathbf{x}^{(n)} + (\epsilon^*(\mathbf{x}^{(n)}, \theta, \mathbf{m}^{(n)}));\theta), y^{(n)}\right) \right\}$$

**IBP-Ex**: Certified robustness approaches, on the other hand, minimize a certifiable upper-bound of the second term. A class of certifiable approaches known as interval bound propagation methods (IBP) (Mirman et al., 2018; Gowal et al., 2018) propagate input intervals to function value intervals that are guaranteed to contain true function values for any input in the input interval.

We define an input interval for $\mathbf{x}^{(n)}$ as $[\mathbf{x}^{(n)} - \kappa\mathbf{m}^{(n)}, \mathbf{x}^{(n)} + \kappa\mathbf{m}^{(n)}]$ where $\kappa$ is defined in Eqn. 2. We then use bound propagation techniques to obtain function value intervals for the corresponding input interval: $\mathbf{l}^{(n)}, \mathbf{u}^{(n)}$, which are ranges over class logits. Since we wish to train a model that correctly classifies an example irrespective of the value of the irrelevant features, we wish to maximize the minimum probability assigned to the correct class, which is obtained by combining minimum logit for the correct class with maximum logit for incorrect class: $\tilde{f}(\mathbf{x}^{(n)}, y^{(n)}, \mathbf{l}^{(n)}, \mathbf{u}^{(n)}; \theta) \triangleq \mathbf{l}^{(n)} \odot \bar{\mathbf{y}}^{(n)} + \mathbf{u}^{(n)} \odot (\mathbf{1} - \bar{\mathbf{y}}^{(n)})$ where $\bar{\mathbf{y}}^{(n)} \in \{0,1\}^c$ denotes the one-hot transformation of the label $y^{(n)}$ into a c-length vector for $c$ classes. We refer to this version of the loss as IBP-Ex, summarized

below.

$$\mathbf{l}^{(n)}, \mathbf{u}^{(n)} = IBP(f(\bullet; \theta), [\mathbf{x}^{(n)} - \kappa \times \mathbf{m}^{(n)}, \mathbf{x}^{(n)} + \kappa \times \mathbf{m}^{(n)}])$$
$$\tilde{f}(\mathbf{x}^{(n)}, y^{(n)}, \mathbf{l}^{(n)}, \mathbf{u}^{(n)}; \theta) \triangleq \mathbf{l}^{(n)} \odot \bar{\mathbf{y}}^{(n)} + \mathbf{u}^{(n)} \odot (\mathbf{1} - \bar{\mathbf{y}}^{(n)})$$
$$\theta^* = \arg\min_\theta \sum_n \ell\left(f(\mathbf{x}^{(n)}; \theta), y^{(n)}\right) + \alpha\ell\left(\tilde{f}(\mathbf{x}^{(n)}, y^{(n)}, \mathbf{l}, \mathbf{u}; \theta), y^{(n)}\right) \quad (3)$$

Despite its computational efficiency, IBP is known to suffer from scaling issues when the model is too big. Consequently, it is better to use IBP-Ex only when the model is small (less than four layers of CNN or feed-forward) and if computational efficiency is desired. On the other hand, we do not anticipate any scaling issues when using PGD-Ex.

**Combined robustness and regularization**: PGD-Ex+Grad-Reg, IBP-Ex+Grad-Reg. We combine robustness and regularization by simply combining their respective loss terms. We show the objective for IBP-Ex+Grad-Reg below, PGD-Ex+Grad-Reg follows similarly.

$$\theta^* = \arg\min_\theta \sum_n \ell\left(f(\mathbf{x}^{(n)}; \theta), y^{(n)}\right) + \alpha\ell\left(\tilde{f}(\mathbf{x}^{(n)}, y^{(n)}, \mathbf{l}, \mathbf{u}; \theta), y^{(n)}\right) + \lambda\mathcal{R}(\theta). \quad (4)$$

$\lambda\mathcal{R}(\theta)$ and $\alpha, \tilde{f}$ are as defined in Eqn. 5 and Eqn. 3 respectively. In Section 4, 5.2, we demonstrate the complementary strengths of robustness and regularization-based methods.

## 4 Theoretical Motivation

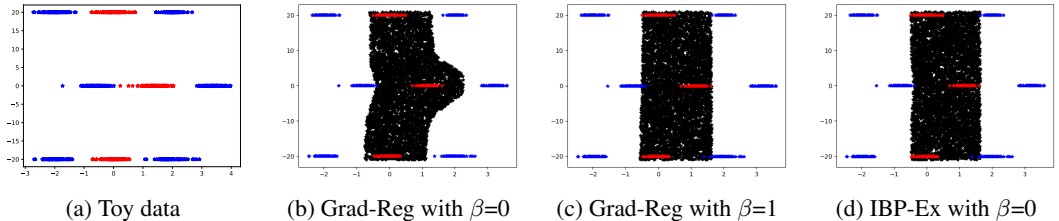

(a) Toy data     (b) Grad-Reg with $\beta=0$     (c) Grad-Reg with $\beta=1$     (d) IBP-Ex with $\beta=0$

Figure 1: Illustration of the uneasy relationship between Grad-Reg and smoothing strength. (b) The decision boundary is nearly vertical (zero gradient wrt to nuisance y-axis value) for all training points and yet varies as a function of y value when fitted using Grad-Reg with $\beta = 0$. (c) Grad-Reg requires strong model smoothing ($\beta = 1$) in order to translate local insensitivity to global robustness to y-coordinate. (d) IBP-Ex, on the other hand, fits vertical pair of lines without any model smoothing.

In this section, we motivate the merits and drawbacks of robustness-based over regularization-based methods. Through non-parametric analysis in Theorems 1, 2, we argue that (a) regularization methods are robust to perturbations of irrelevant features (identified by the mask) only when the underlying model is sufficiently smoothed, thereby potentially compromising performance, (b) robust training upper-bounds deviation in function values when irrelevant features are perturbed, which can be further suppressed by using a more effective robust training. Although our analysis is restricted to nonparametric models for the ease of analysis, we empirically verify our claims with parametric neural network optimized using a gradient-based optimizer. We then highlight a limitation of robustness-based methods when the number of irrelevant features is large through Proposition 1.

### 4.1 Merits of Robustness-based methods

Consider a two-dimensional regression task, i.e. $\mathbf{x}^{(n)} \in \mathcal{X}$ and $y \in \mathbb{R}$. Assume that the second feature is the shortcut that the model should not use for prediction, and denote by $\mathbf{x}_j^{(n)}$ the $j^{th}$ dimension of $n^{th}$ point. We infer a regression function $f$ from a Gaussian process prior $f \sim GP(f; 0, K)$ with a squared exponential kernel where $k(x, \tilde{x}) = \exp(-\sum_i \frac{1}{2} \frac{(x_i - \tilde{x}_i)^2}{\theta_i^2})$. As a result, we have two hyperparameters $\theta_1, \theta_2$, which are length scale parameters for the first and second dimensions respectively. Further, we impose a Gamma prior over the length scale: $\theta_i^{-2} \sim \mathcal{G}(\alpha, \beta)$.

**Theorem 1** (Grad-Reg). *We infer a regression function $f$ from a GP prior as described above with the additional supervision of $[\partial f(\mathbf{x})/\partial x_2]|_{\mathbf{x}^{(i)}} = 0, \quad \forall i \in [1, N]$. Then the function value deviations to perturbations on irrelevant feature are lower bounded by a value proportional to the perturbation strength $\delta$ as shown below.*

$$f(\mathbf{x} + [0, \delta]^T) - f(\mathbf{x}) \geq \frac{2\delta\alpha}{\beta}\Theta(x_1^2 x_2^6 + \delta x_1^2 x_2^5) \tag{5}$$

Full proof of Theorem 1 is in Appendix A, we provide the proof outline below.

*Proof sketch.* We exploit the flexibility of GPs to accommodate observations on transformed function values if the transformation is closed under linear operation (Hennig et al., 2022) (see Chapter 4). Since gradient is a linear operation (contributors, 2022), we simply revise the observations $y$ to include the $N$ partial derivative observations for each training point with appropriately defined kernel. We then express the inferred function, which is the posterior $f(\mathbf{x} \mid \mathcal{D}_T)$ in closed form.

We then derive a bound on function value deviations to perturbation on the second feature and simplify using simple arithmetic and Bernoulli's inequality to derive the final expression. □

We observe from Theorem 1 that if we wish to infer a function that is robust to irrelevant feature perturbations, we need to set $\frac{\alpha}{\beta}$ to a very small value. Since the expectation of gamma distributed inverse-square length parameter is $\mathbb{E}[\theta^{-2}] = \frac{\alpha}{\beta}$, which we wish to set very small, we are, in effect, sampling functions with very large length scale parameter i.e. strongly smooth functions. This result brings us to the intuitive takeaway that regularization using Grad-Reg applies globally only when the underlying family of functions is sufficiently smooth. The result may also hold for any other local-interpretation methods that is closed under linear operation. One could also argue that we can simply use different priors for different dimensions, which would resolve the over-smoothing issue. However, we do not have access to parameters specific to each dimension in practice and especially with DNNs, therefore only overall smoothness may be imposed such as with parameter norm regularization in Eqn. 1.

We now look at properties of a function fitted using robustness methods and argue that they bound deviations in function values better. In order to express the bounds, we introduce a numerical quantity called coverage (C) to measure the effectiveness of a robust training method. We first define a notion of inputs covered by a robust training method as $\hat{\mathcal{X}} \triangleq \{\mathbf{x} \mid \mathbf{x} \in \mathcal{X}, \ell(f(\mathbf{x}; \theta), y) < \phi\} \subset \mathcal{X}$ for a small positive threshold $\phi$ on loss. We define coverage as the maximum distance along second coordinate between any point in $\mathcal{X}$ and its closest point in $\hat{\mathcal{X}}$, i.e. $C \triangleq \max_{\mathbf{x} \in \mathcal{X}} \min_{\mathbf{x} \in \hat{\mathcal{X}}} |\mathbf{x}_2 - \hat{\mathbf{x}}_2|$. We observe that C is small if the robust training is effective. In the extreme case when training minimizes the loss for all points in the input domain, i.e. $\hat{\mathcal{X}} = \mathcal{X}$, then C=0.

**Theorem 2.** *When we use a robustness algorithm to regularize the network, the fitted function has the following property.*

$$|f(\mathbf{x} + [0, \delta]^T) - f(\mathbf{x})| \leq 2C\frac{\alpha}{\beta}\delta_{max} f_{max}. \tag{6}$$

$\delta_{max}$ *and* $f_{max}$ *are maximum values of* $\Delta x_2$ *and* $f(\mathbf{x})$ *in the input domain* $(\mathcal{X})$ *respectively.*

*Proof sketch.* We begin by estimating the function deviation between an arbitrary point and a training instance that only differ on $x_2$, which is bounded by the product of maximum slope $(\max_{\mathbf{x}} \partial f(\mathbf{x})/\partial x_2)$ and $\Delta x_2$. We then estimate the maximum slope (lipschitz constant) for a function inferred from GP. Finally, function deviation between two arbitrary points is twice the maximum deviation between an arbitrary and a training instance that is estimated in the first step. □

Full proof is in Appendix B. The statement shows that deviations in function values are upper bounded by a factor proportional to $C$, which can be dampened by employing an effective robust training method. We can therefore control the deviations in function values without needing to regress $\frac{\alpha}{\beta}$ (i.e. without over-smoothing).

**Further remarks on sources of over-smoothing in regularization-based methods.** We empirically observed that the term $\mathcal{R}(\theta)$ (of Eqn. 1), which supervises explanations, also has a smoothing effect on the model when the importance scores (IS) are not well normalized, which is often the case. This is because reducing IS($\mathbf{x}$) everywhere will also reduce saliency of irrelevant features.

**Empirical verification with a toy dataset.** For empirical verification of our results, we fit a 3-layer feed-forward network on a two-dimensional data shown in Figure 1 (a), where color indicates the label. We consider fitting a model that is robust to changes in the second feature shown on y-axis. In Figures 1 (b), (c), we show the Grad-Reg fitted classifier using gradient ($\partial f / \partial x_2$ for our case) regularization for two different strengths of parameter smoothing (0 and 1 respectively). With weak smoothing, we observe that the fitted classifier is locally vertical (zero gradient along y-axis), but curved overall (Figure 1 (b)), which is fixed with strong smoothing (Figure 1 (c)). On the other hand, IBP-Ex fitted classifier is nearly vertical without any parameter regularization as shown in (d). This example illustrates the need for strong model smoothing when using a regularization-based method.

## 4.2 Drawbacks of Robustness-based methods

Although robust training is appealing in low dimensions, their merits do not transfer well when the space of irrelevant features is high-dimensional owing to difficulty in solving the inner maximization of Eqn. 2. Sub-par estimation of the maximization term may learn parameters that still depend on the irrelevant features. We demonstrate this below with a simple exercise.

**Proposition 1.** *Consider a regression task with $D + 1$-dimensional inputs $\mathbf{x}$ where the first $D$ dimensions are irrelevant, and assume they are $x_d = y, d \in [1, D]$ while $x_{D+1} \sim \mathcal{N}(y, 1/K)$. The MAP estimate of linear regression parameters $f(\mathbf{x}) = \sum_{d=1}^{D+1} w_d x_d$ when fitted using Avg-Ex are as follows: $w_d = 1/(D + K), \quad d \in [1, D]$ and $w_{D+1} = K/(K + D)$.*

We present the proof in Appendix C. We observe that as D increases, the weight of the only relevant feature ($x_{D+1}$) diminishes. On the other hand, the weight of the average feature: $\frac{1}{D} \sum_{d=1}^{D} x_d$ , which is $D/(D + K)$ approaches 1 as $D$ increases. This simple exercise demonstrates curse of dimensionality for robustness-based methods. For this reason, we saw major empirical gains when combining robustness methods with a regularization method especially when the number of irrelevant features is large such as in the case of Decoy-MNIST dataset, which is described in the next section.

## 5 Experiments

We evaluate different methods on four datasets: one synthetic and three real-world. The synthetic dataset is similar to decoy-MNIST of Ross et al. (2017) with induced shortcuts and is presented in Section 5.2. For evaluation on practical tasks, we evaluated on a plant phenotyping (Shao et al., 2021) task in Section 5.3, skin cancer detection (Rieger et al., 2020) task presented in Section 5.4, and object classification task presented in Section 5.5. All the datasets contain a known spurious feature, and were used in the past for evaluation of MLX methods. Figure 2 summarises the three datasets, notice that we additionally require in the training dataset the specification of a mask identifying irrelevant features of the input; the patch for ISIC dataset, background for plant dataset, decoy half for Decoy-MNIST images, and label-specific irrelevant region approved by humans for Salient-Imagenet.

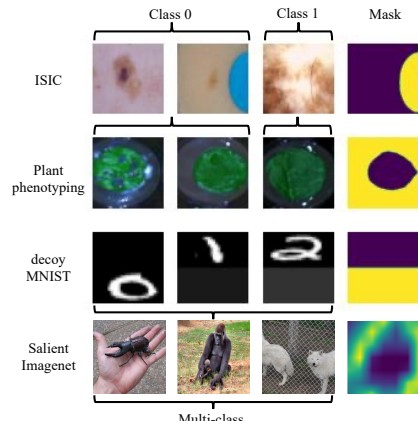

Figure 2: Sample images and masks for different datasets. The shown mask corresponds to the image from the second column.

### 5.1 Setup

#### 5.1.1 Baselines

We denote by ERM the simple minimization of cross-entropy loss (using only the first loss term of Equation 1). We also compare with G-DRO(Sagawa et al., 2019), which also has the objective of avoiding to learn known irrelevant features but is supervised through group label (see Section 6). Although the comparison is unfair toward G-DRO because MLX methods use richer supervision of per-example masks, it serves as a baseline that can be slightly better than ERM in some cases.

**Regulaization-based methods.** Grad-Reg and CDEP, which were discussed in Section 2. We omit comparison with Shao et al. (2021) because their code is not publicly available and is non-trivial to implement the influence-function based regularization.

**Robustness-based methods.** Avg-Ex, PGD-Ex, IBP-Ex along with **combined robustness and regularization methods**. IBP-Ex+Grad-Reg, PGD-Ex+Grad-Reg that are described in Section 3.

### 5.1.2  Metrics

Since two of our datasets are highly skewed (Plant and ISIC), we report (macro-averaged over labels) accuracy denoted "Avg Acc" and worst accuracy "Wg Acc" over groups of examples with groups appropriately defined as described below. On Salient-Imagenet, we report using average accuracy and Relative Core Sentivity (RCS) following the standard practice.

**Wg Acc.** Worst accuracy among groups where groups are appropriately defined. Different labels define the groups for decoy-MNIST and plant dataset, which therefore have ten and two groups respectively. In ISIC dataset, different groups are defined by the cross-product of label and presence or absence of the patch. We denote this metric as "Wg Acc", which is a standard metric when evaluating on datasets with shortcut features (Sagawa et al., 2019).

**RCS** proposed in Singla et al. (2022) was used to evaluate models on Salient-Imagenet. The metric measures the sensitivity of a model to noise in spurious and core regions. High RCS implies low dependence on irrelevant regions, and therefore desired. Please see Appendix E.2 for more details.

### 5.1.3  Training and Implementation details

**Choice of the best model.** We picked the best model using the held-out validation data. We then report the performance on test data averaged over three seeds corresponding to the best hyperparameter.

**Network details.** We use four-layer CNN followed by three-fully connected layers for binary classification on ISIC and plant dataset following the setting in Zhang et al. (2019), and three-fully connected layers for multi classification on decoy-MNIST dataset. Pretrained Resnet-18 is used as the backbone model for Salient-Imagenet.

More details about network architecture, datasets, data splits, computing specs, and hyperparameters can be found in Appendix E.

| Dataset → | Decoy-MNIST | | Plant | | ISIC | | Salient-Imagenet | |
|---|---|---|---|---|---|---|---|---|
| Method↓ | Avg Acc | Wg Acc | Avg Acc | Wg Acc | Avg Acc | Wg Acc | Avg Acc | RCS |
| ERM | 15.1 | 10.5 | 71.3 | 54.8 | 77.3 | 55.9 | 96.4 | 47.9 |
| G-DRO | 64.1 | 28.1 | 74.2 | 58.0 | 66.6 | 58.5 | - | - |
| Grad-Reg | 72.5 | 46.2 | 72.4 | 68.2 | 76.4 | 60.2 | 88.3 | 52.5 |
| CDEP | 14.5 | 10.0 | 67.9 | 54.2 | 73.4 | 60.9 | - | - |
| Avg-Ex | 29.5 | 19.5 | 76.3 | 64.5 | 77.1 | 55.2 | - | - |
| PGD-Ex | 67.6 | 51.4 | 79.8 | 78.5 | **78.7** | **64.4** | 93.8 | 58.7 |
| IBP-Ex | 68.1 | 47.6 | 76.6 | 73.8 | 75.1 | 64.2 | - | - |
| P+G | **96.9** | **95.8** | 79.4 | 76.7 | **79.6** | 67.5 | **94.6** | **65.0** |
| I+G | **96.9** | **95.0** | **81.7** | **80.1** | 78.4 | 65.2 | - | - |

Table 1: Macro-averaged (Avg) accuracy and worst group (Wg) accuracy on (a) decoy-MNIST, (b) plant dataset, (c) ISIC dataset along with relative core sensitivity (RCS) metric on (d) Salient-Imagenet. Statistically significant numbers are marked in bold. Please see Table 4 in Appendix for standard deviations. I+G is short for IBP-Ex+Grad-Reg and P+G for PGD-Ex+Grad-Reg.

### 5.2  Decoy-MNIST

Decoy-MNIST dataset is inspired from MNIST-CIFAR dataset of Shah et al. (2020) where a very simple label-revealing color based feature (decoy) is juxtaposed with a more complex feature (MNIST image) as shown in Figure 1. We also randomly swap the position of decoy and MNIST parts, which makes ignoring the decoy part more challenging. We then validate and test on images where decoy part is set to correspond with random other label. Note that our setting is different from decoy-mnist of CDEP (Rieger et al., 2020), which is likely why we observed discrepancy in results reported in our paper and CDEP (Rieger et al., 2020). We further elaborate on these differences in Appendix G.

We make the following observations from Decoy-MNIST results presented in Table 1. ERM is only slightly better than a random classifier confirming the simplicity bias observed in the past (Shah et al., 2020). Grad-Reg, PGD-Ex and IBP-Ex perform comparably and better than ERM, but when combined (IBP-Ex+Grad-Reg,PGD-Ex+Grad-Reg) they far exceed their individual performances.

In order to understand the surprising gains when combining regularization and robustness methods, we draw insights from gradient explanations on images from train split for Grad-Reg and IBP-Ex. We looked at $s_1 = \mathcal{M}\left[\left\|\mathbf{m}^{(n)} \times \frac{\partial f(\mathbf{x}^{(n)})}{\partial \mathbf{x}^{(n)}}\right\|\right]$ and $s_2 = \mathcal{M}\left[\left\|\mathbf{m}^{(n)} \times \frac{\partial f(\mathbf{x}^{(n)})}{\partial \mathbf{x}^{(n)}}\right\| \Big/ \left\|(\mathbf{1} - \mathbf{m}^{(n)}) \times \frac{\partial f(\mathbf{x}^{(n)})}{\partial \mathbf{x}^{(n)}}\right\|\right]$, where $\mathcal{M}[\bullet]$ is median over all the examples and $\|\cdot\|$ is $\ell_2$-norm. For an effective algorithm, we expect both $s_1, s_2$ to be close to zero. However, the values of $s_1, s_2$ is 2.3e-3, 0.26 for the best model fitted using Grad-Reg and 6.7, 0.05 for IBP-Ex. We observe that Grad-Reg has lower $s_1$ while IBP-Ex has lower $s_2$, which shows that Grad-Reg is good at dampening the contribution of decoy part but also dampened contribution of non-decoy likely due to over-smoothing. IBP-Ex improves the contribution of the non-decoy part but did not fully dampen the decoy part likely because high dimensional space of irrelevant features, i.e. half the image is irrelevant and every pixel of irrelevant region is indicative of the label. When combined, IBP-Ex+Grad-Reg has low $s_1, s_2$, which explains the increased performance when they are combined.

### 5.3 Plant Phenotyping

Plant phenotyping is a real-world task of classifying images of a plant leaf as healthy or unhealthy. About half of leaf images are infected with a Cercospora Leaf Spot (CLS), which are the black spots on leaves as shown in the first image in the second row of Figure 2. Schramowski et al. (2020) discovered that standard models exploited unrelated features from the nutritional solution in the background in which the leaf is placed, thereby performing poorly when evaluated outside of the laboratory setting. Thus, we aim to regulate the model not to focus on the background of the leaf using binary specification masks indicating where the background is located. Due to lack of out-of-distribution test set, we evaluate with in-domain test images but with background pixels replaced by a constant pixel value, which is obtained by averaging over all pixels and images in the training set. We replace with an average pixel value in order to avoid any undesired confounding from shifts in pixel value distribution. In Appendix F.5, we present accuracy results when adding varying magnitude of noise to the background, which agree with observations we made in this section. More detailed analysis of the dataset can be found in Schramowski et al. (2020).

Table 1 contrasts different algorithms on the plant dataset. All the algorithms except CDEP improve over ERM, which is unsurprising given our test data construction; any algorithm that can divert focus from the background pixels can perform well. Wg accuracy of robustness (except Avg-Ex) and combined methods far exceed any other method by 5-12% over the next best baseline and by 19-26% accuracy point over ERM. Surprisingly, even Avg-Ex has significantly improved the performance over ERM likely because spurious features in the background are spiky or unstable, which vanishes with simple perturbations.

We visualize the interpretations of models obtained using SmoothGrad (Smilkov et al., 2017) trained with five different methods for three sample images from the train split in Figure 3. As expected, ERM has strong dependence on

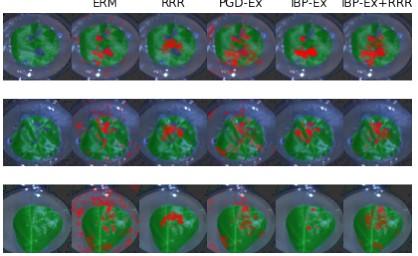

Figure 3: Visual heatmap of salient features for different algorithms on three sample train images of Plant data. Importance score computed with Smooth-Grad (Smilkov et al., 2017).

non-leaf background features. Although Grad-Reg features are all on the leaf, they appear to be localized to a small region on the leaf, which is likely due to over-smoothing effect of its loss. IBP-Ex, IBP-Ex+Grad-Reg on the other hand draws features from a wider region and has more diverse pattern of active pixels.

### 5.4 ISIC: Skin Cancer Detection

ISIC is a dataset of skin lesion images, which are to be classified cancerous or non-cancerous. Since half the non-cancerous images in the dataset contains a colorful patch as shown in Figure 2, standard

DNN models depend on the presence of a patch for classification while compromising the accuracy on non-cancerous images without a patch (Codella et al., 2019; Tschandl et al., 2018). We follow the standard setup and dataset released by Rieger et al. (2020), which include masks highlighting the patch. Notably, (Rieger et al., 2020) employed a different evaluation metric, F1, and a different architecture — a VGG model pre-trained on ImageNet. This may explain the worse-than-expected performance of CDEP from Rieger et al. (2020). More details are available in Appendix G.

Traditionally, three groups are identified in the dataset: non-cancerous images without patch (NCNP) and with patch (NCP), and cancerous images (C). In Table 2, we report on per-group accuracies for different algorithms. Detailed results with error bars are shown in Table 5 of Appendix F. The Wg accuracy (of Table 1) may not match with the worst of the average group accuracies in Table 2 because we report average of worst accuracies.

We now make the following observations. ERM performs the worst on the NPNC group confirming that predictions made by a standard model depend on the patch. The accuracy on the PNC group is high overall perhaps because PNC group images that are consistently at a lower scale (see middle column of Figure 2 for an example) are systematically more easier to classify even when the patch is not used for classification. Although human-explanations for this dataset, which only identifies the patch if present, do not full specify all spurious correlations, we still saw gains when learning from them. Grad-Reg and CDEP improved NPNC accuracy at the expense of C's accuracy while still performing relatively poor on Wg accuracy. Avg-Ex performed no better

| Method | NPNC | PNC | C |
|---|---|---|---|
| ERM | 55.9 | 96.5 | 79.6 |
| Grad-Reg | 67.1 | 99.0 | 63.2 |
| CDEP | 72.1 | 98.9 | 62.2 |
| Avg-Ex | 62.3 | 97.8 | 71.0 |
| PGD-Ex | 65.4 | 99.0 | 71.7 |
| IBP-Ex | 68.4 | 98.5 | 67.7 |
| I+G | 66.6 | 99.6 | 68.9 |
| P+G | 69.6 | 98.8 | 70.4 |

Table 2: Per-group accuracies on ISIC.

than ERM whereas PGD-Ex, IBP-Ex, IBP-Ex+Grad-Reg, and PGD-Ex+Grad-Reg significantly improved Wg accuracy over other baselines. The reduced accuracy gap between NPNC and C when using combined methods is indicative of reduced dependence on patch. We provide a detailed comparison of PGD-Ex and IBP-Ex in Appendix F.3

## 5.5 Salient-Imagenet

Salient-Imagenet (Singla & Feizi, 2021; Singla et al., 2022) is a large scale dataset based on Imagenet with pixel-level annotations of irrelevant (spurious) and relevant (core) regions. Salient-Imagenet is an exemplary effort on how explanation masks can be obtained at scale. The dataset contains 232 classes and 52,521 images (with about 226 examples per class) along with their core and spurious masks. We made a smaller evaluation subset using six classes with around 600 examples. Each of the six classes contain at least one spurious feature identified by the annotator. Please refer to Appendix E.4 for more details. We then tested different methods for their effectiveness in learning a model that ignores the irrelevant region. We use a ResNet-18 model pretrained on ImageNet as the initialization.

In Table 1, we report RCS along with overall accuracy. Appendix F.2 contains further results including accuracy when noise is added to spurious (irrelevant) region. The results again substantiate the relative strength of robustness-methods over regularization-based. Furthermore, we highlight the considerable improvement offered by the novel combination of robustness-based and regularization-based methods.

## 5.6 Overall results

Among the regularization-based methods, Grad-Reg performed the best while also being simple and intuitive.

Robustness-based methods except Avg-Ex are consistently and effortlessly better or comparable to regularization-based methods on all the benchmarks with an improvement to Wg accuracy by 3-10% on the two real-world datasets. Combined methods are better than their constituents on all the datasets readily without much hyperparameter tuning.

In Appendix, we have additional experiment results; generality to new explanation methods (Appendix F.6) and new attention map-based architecture (Appendix F.7), and sensitivity to hyperparameters of PGD-Ex on Plant and ISIC dataset (Appendix F.4) and other perturbation-based methods on Decoy-MNIST dataset (Appendix F.8).

## 6 Related Work

**Sub-population shift robustness.** Sagawa et al. (2019) popularized a problem setting to avoid learning known spurious correlation by exploiting group labels, which are labels that identify the group an example. Unlike MLX, the sub-population shift problem assumes that training data contains groups of examples with varying strengths of spurious correlations. For example in the ISIC dataset, images with and without patch make different groups. Broadly, different solutions attempt to learn a hypothesis that performs equally well on all the groups via importance up-weighting (Shimodaira (2000); Byrd & Lipton (2019)), balancing subgroups(Cui et al. (2019); Izmailov et al. (2022); Kirichenko et al. (2022)) or group-wise robust optimization (Sagawa et al. (2019)). Despite the similarities, MLX and sub-population shift problem setting have some critical differences. Sub-population shift setting assumes sufficient representation of examples from groups with varying levels of spurious feature correlations. This assumption may not always be practical, e.g. in medical imaging, which is when MLX problem setting may be preferred.

**Learning from human-provided explanations.** Learning from explanations attempts to ensure that model predictions are "right for the right reasons" (Ross et al., 2017). Since gradient-based explanations employed by Ross et al. (2017) are known to be unfaithful (Murphy, 2023) (Chapter 33) (Wicker et al., 2022), subsequent works have proposed to replace the explanation method while the overall loss structure remained similar. Shao et al. (2021) proposed to regularize using an influence function, while Rieger et al. (2020) proposed to appropriately regularize respective contributions of relevant or irrelevant features to the classification probability through an explanation method proposed by Singh et al. (2018). In a slight departure from these methods, Selvaraju et al. (2019) used a loss objective that penalises ranking inconsistencies between human and the model's salient regions demonstrated for VQA applications. Stammer et al. (2021) argued for going beyond pixel-level importance scores to concept-level scores for the ease of human intervention. On the other hand, Singla et al. (2022) studied performance when augmenting with simple perturbations of irrelevant features and with the gradient regularization of Ross et al. (2017). This is the only work, to the best of our knowledge, that explored robustness to perturbations for learning from explanations.

**Robust training.** Adversarial examples were first popularized for neural networks by Szegedy et al. (2013), and have been a significant issue for machine learning models for at least a decade (Biggio & Roli, 2018). Local methods for computing adversarial attacks have been studied (Madry et al., 2017), but it is well known that adaptive attacks are stronger (i.e., more readily fool NNs) than general attack methods such as PGD (Tramer et al., 2020). Certification approaches on the other hand are guaranteed to be worse than any possible attack (Mirman et al., 2018), and training with certification approaches such as IBP have been found to provide state-of-the-art results in terms of provable robustness (Gowal et al., 2018), uncertainty (Wicker et al., 2021), explainability (Wicker et al., 2022), and fairness (Benussi et al., 2022).

## 7 Conclusions

By casting MLX as a robustness problem and using human explanations to specify the manifold of perturbations, we have shown that it is possible to alleviate the need for strong parameter smoothing of earlier approaches. Borrowing from the well-studied topic of robustness, we evaluated two strong approaches, one from adversarial robustness (PGD-Ex) and one from certified robustness (IBP-Ex). In our evaluation spanning seven methods and four datasets including two real-world datasets we found that PGD-Ex and IBP-Ex performed better than any previous approach, while our final proposal IBP-Ex+Grad-Reg and PGD-Ex+Grad-Reg of combining IBP-Ex and PGD-Ex with a light-weight interpretation based method respectively have consistently performed the best without compromising compute efficiency by much.

**Limitations.** Detecting and specifying irrelevant regions per-example by humans is a laborious and non-trivial task. Hence, it is interesting to see the effects of learning from incomplete explanations, which we leave for future work. Although robustness-based methods are very effective as we demonstrated, they can be computationally expensive (PGD-Ex) or scale poorly to large networks (IBP-Ex).

**Broader Impact.** Our work is a step toward reliable machine learning intended for improving ML for all. To the best of our knowledge, we do not foresee any negative societal impacts.

## 8 Acknowledgements

MW acknowledges support from Accenture. MW and AW acknowledge support from EPSRC grant EP/V056883/1. AW acknowledges support from a Turing AI Fellowship under grant EP/V025279/1, and the Leverhulme Trust via CFI. We thank Pingfan Song, Yongchao Huang, Usman Anwar and multiple anonymous reviewers for engaging in thoughtful discussions, which helped greatly improve the quality of our paper.

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

# Supporting material for "Use Perturbations when Learning from Explanations"

## A   Proof of Theorem 1

We restate the result of Theorem 1 for clarity.

We infer a regression function $f$ from a GP prior as described above with the additional supervision of $[\partial f(\mathbf{x})/\partial x_2]|_{\mathbf{x}^{(i)}} = 0, \quad \forall i \in [1, N]$. Then the function value deviations to perturbations on irrelevant feature are lower bounded by a value proportional to the perturbation strength $\delta$ as shown below.

$$f(\mathbf{x} + [0, \delta]^T) - f(\mathbf{x}) \geq \frac{2\delta\alpha}{\beta}\Theta(x_1^2 x_2^6 + \delta x_1^2 x_2^5)$$

**Proof outline.** We first show that the posterior mean of the function estimates marginalised over hyperparameters with Gamma prior has the following closed form where $d(x, y) = (x - y)^2/2$ and $\tilde{y}$ denotes original observations $y$ augmented with observations on gradients, which is described in more detail further below.

$$f(x) \triangleq \mathbb{E}_\theta[m_x] = \int \int m_x \mathcal{G}(\theta_1^{-2}; \alpha, \beta)\mathcal{G}(\theta_2^{-2}; \alpha, \beta)d\theta_1^{-2}d\theta_2^{-2}$$

$$f(\mathbf{x}) = \sum_{n=1}^N \left(\frac{1}{1 + \frac{d(x_1, x_1^{(n)})}{\beta}}\right)^\alpha \left(\frac{1}{1 + \frac{d(x_2, x_2^{(n)})}{\beta}}\right)^\alpha \left[\tilde{y}^{(n)} + \frac{\frac{\alpha}{\beta}(x_2 - x_2^{(n)})}{1 + \frac{d(x_2, x_2^{(n)})}{\beta}}\tilde{y}^{(n+N)}\right]$$

We then derive the following lower bound on the function value deviations and finally use simple inequalities to arrive at the final result.

$$f(\mathbf{x} + [0, \delta]^T]) - f(\mathbf{x}) \geq \frac{2\delta\alpha}{\beta}\sum_n \left(\frac{1}{1 + \frac{d(x_1, x_1^{(n)})}{\beta}}\right)^\alpha \left(\frac{1}{1 + \frac{d(x_2, x_2^{(n)})}{\beta}}\right)^{\alpha+1}$$
$$\left[(\alpha + 1)\tilde{y}_{n+N}\left(\frac{2(x_2 - x_2^{(n)})[x_2 + \delta - x_2^{(n)}]}{\beta + d(x_2, x_2^{(n)})} - 1\right) - \tilde{y}_n\right]$$

*Proof.* We first derive the augmented set of observations $(\hat{y})$ and $\hat{K}$ explained in the main section.

$$\hat{y} = [y_1, y_2, \ldots, y_N, \partial f(\mathbf{x}^{(1)})/\partial x_2, \partial f(\mathbf{x}^{(2)})/\partial x_2, \ldots, \partial f(\mathbf{x}^{(N)})/\partial x_2]^T$$

$$k(x^{(i)}, x^{(j)}) = \begin{cases} \exp(-\frac{1}{2}\sum_{k=1}^2 \frac{(x_k^{(i)} - x_k^{(j)})^2}{\theta_k^2}) & \text{when i, j} \leq \text{N} \\ \frac{(x_2^{(i)} - x_2^{(j)})}{\theta_2^2}\exp(-\frac{1}{2}\sum_{k=1}^2 \frac{(x_k^{(i)} - x_k^{(j)})^2}{\theta_k^2}) & \text{when i} \leq \text{N, j>N} \\ -\frac{(x_2^{(i)} - x_2^{(j)})}{\theta_2^2}\exp(-\frac{1}{2}\sum_{k=1}^2 \frac{(x_k^{(i)} - x_k^{(j)})^2}{\theta_k^2}) & \text{when j} \leq \text{N, i>N} \\ -2\frac{(x_2^{(i)} - x_2^{(j)})^2}{\theta_2^4}\exp(-\frac{1}{2}\sum_{k=1}^2 \frac{(x_k^{(i)} - x_k^{(j)})^2}{\theta_k^2}) & \\ +\frac{1}{\theta_2^2}\exp(-\frac{1}{2}\sum_{k=1}^2 \frac{(x_k^{(i)} - x_k^{(j)})^2}{\theta_k^2}) & \text{when i, j > N} \end{cases}$$

These results follow directly from the results on covariance between observations of f and its partial derivative below (Hennig et al., 2022).

$$\text{cov}(f(x), \frac{\partial f(\tilde{x})}{\partial \tilde{x}}) = \frac{\partial k(x, \tilde{x})}{\partial \tilde{x}}$$
$$\text{cov}(\frac{\partial f(x)}{x}, \frac{\partial f(\tilde{x})}{\tilde{x}}) = \frac{\partial^2 k(x, \tilde{x})}{\partial x \partial \tilde{x}}$$

The posterior value of the function at an arbitrary point $\mathbf{x}$ would then be of the form $p(f(\mathbf{x}) \mid \mathcal{D}) \sim \mathcal{N}(f(\mathbf{x}); m_x, k_x)$ where $m_x$ and $k_x$ are have the following closed form for Gaussian prior and

Gaussian likelihood in our case.

$$m_x = k(x, X)K_{XX}^{-1}\hat{y}$$
$$k_x = k(x, x) - k(x, X)K_{XX}^{-1}k(X, x)$$

Since $m_x, k_x$ are functions of the parameters $\theta_1, \theta_2$, we obtain the closed form for posterior mean by imposing a Gamma prior over the two parameters. For brevity, we denote by $d(x, \tilde{x}) = (x - \tilde{x})^2/2$ and $\tilde{y}^{(i)}$ is the $i^{(th)}$ component of $\hat{K}_{XX}^{-1}\hat{y}$.

$$f(x) \triangleq \mathbb{E}_\theta[m_x] = \int\int m_x \mathcal{G}(\theta_1^{-2}; \alpha, \beta)\mathcal{G}(\theta_2^{-2}; \alpha, \beta)d\theta_1^{-2}d\theta_2^{-2}$$

$$= \int\int \left[\sum_{n=1}^N k(x, x^{(n)})\tilde{y}_n + \sum_{n=1}^N \frac{(x_2 - x_2^{(n)})}{\theta_2^2}k(x, x^{(n)})\tilde{y}_{n+N}\right]\mathcal{G}(\theta_1^{-2}; \alpha, \beta)\mathcal{G}(\theta_2^{-2}; \alpha, \beta)d\theta_1^{-2}d\theta_2^{-2}$$

$$\int\int k(\mathbf{x}, \mathbf{x}^{(n)})\tilde{y}_n\mathcal{G}(\theta_1^{-2}; \alpha, \beta)\mathcal{G}(\theta_2^{-2}; \alpha, \beta)d\theta_1^{-2}d\theta_2^{-2}$$

$$= \int\int \exp\left(-\frac{\theta_1^{-2}(x_1 - x_1^{(n)})^2}{2} + \frac{\theta_2^{-2}(x_2 - x_2^{(n)})^2}{2}\right)\frac{\beta^\alpha}{\Gamma(\alpha)}\theta_1^{-2\alpha+2}\exp\left(-\beta\theta_1^{-2}\right)$$

$$\frac{\beta^\alpha}{\Gamma(\alpha)}\theta_2^{-2\alpha+2}\exp\left(-\beta\theta_2^{-2}\right)\tilde{y}_n d\theta_1^{-2}d\theta_2^{-2}$$

$$= \left(\frac{\beta}{\beta + \frac{(x_1 - x_1^{(n)})^2}{2}}\right)^\alpha \left(\frac{\beta}{\beta + \frac{(x_2 - x_2^{(n)})^2}{2}}\right)^\alpha \tilde{y}_n$$

$$\int\int \frac{x_2 - x_2^{(n)}}{\theta_2^2}k(\mathbf{x}, \mathbf{x}^{(n)})\tilde{y}_{n+N}\mathcal{G}(\theta_1^{-2}; \alpha, \beta)\mathcal{G}(\theta_2^{-2}; \alpha, \beta)d\theta_1^{-2}d\theta_2^{-2}$$

$$= (x_2 - x_2^{(n)})\left(\frac{\beta}{\beta + \frac{(x_1 - x_1^{(n)})^2}{2}}\right)^\alpha \frac{\beta^\alpha/\Gamma(\alpha)}{(\beta + \frac{(x_2 - x_2^{(n)})^2}{2})^{\alpha+1}/\Gamma(\alpha+1)}\tilde{y}_{n+N}$$

$$= \left(\frac{\beta}{\beta + \frac{(x_1 - x_1^{(n)})^2}{2}}\right)^\alpha \frac{\alpha(x_2 - x_2^{(n)})}{\beta + \frac{(x_2 - x_2^{(n)})^2}{2}}\left(\frac{\beta}{\beta + \frac{(x_2 - x_2^{(n)})^2}{2}}\right)^\alpha \tilde{y}_{n+N}$$

Overall, we have the following result.

$$f(x) = \sum_{n=1}^N \left(\frac{1}{1 + \frac{d(x_1, x_1^{(n)})}{\beta}}\right)^\alpha \left(\frac{1}{1 + \frac{d(x_2, x_2^{(n)})}{\beta}}\right)^\alpha \left[\tilde{y}_n + \frac{\frac{\alpha}{\beta}(x_2 - x_2^{(n)})}{1 + \frac{d(x_2, x_2^{(n)})}{\beta}}\tilde{y}_{n+N}\right]$$

We now derive the sensitivity to perturbations on the second dimension for $\Delta\mathbf{x} = [0, \delta]^T$.

$$f(\mathbf{x} + \Delta\mathbf{x}) - f(\mathbf{x}) = \sum_{n=1}^N \left(\frac{1}{1 + \frac{d(x_1, x_1^{(n)})}{\beta}}\right)^\alpha \left\{\left[\left(\frac{1}{1 + \frac{d(x_2+\delta, x_2^{(n)})}{\beta}}\right)^\alpha - \left(\frac{1}{1 + \frac{d(x_2, x_2^{(n)})}{\beta}}\right)^\alpha\right]\tilde{y}_n\right.$$

$$\left.\left[\frac{\frac{\alpha}{\beta}(x_2 + \delta - x_2^{(n)})}{(1 + \frac{d(x_2+\delta, x_2^{(n)})}{\beta})^{\alpha+1}} - \frac{\frac{\alpha}{\beta}(x_2 - x_2^{(n)})}{(1 + \frac{d(x_2, x_2^{(n)})}{\beta})^{\alpha+1}}\right]\tilde{y}_{n+N}\right\} \tag{7}$$

Using Bernoulli inequality, $(1 + x)^r \geq 1 + rx$ if $r \leq 0$, we derive the following inequalities.

$$\left( \frac{1}{1 + \frac{d(x_2+\delta, x_2^{(n)})}{\beta}} \right)^\alpha - \left( \frac{1}{1 + \frac{d(x_2, x_2^{(n)})}{\beta}} \right)^\alpha$$

$$= \left( \frac{1}{1 + \frac{d(x_2, x_2^{(n)})}{\beta}} \right)^\alpha \left[ \left( \frac{\beta + d(x_2, x_2^{(n)})}{\beta + d(x_2 + \delta, x_2^{(n)})} \right)^\alpha - 1 \right]$$

$$\geq \left( \frac{1}{1 + \frac{d(x_2, x_2^{(n)})}{\beta}} \right)^\alpha - \alpha \left[ \frac{\beta + d(x_2 + \delta, x_2^{(n)})}{\beta + d(x_2, x_2^{(n)})} - 1 \right]$$

$$= \left( \frac{1}{1 + \frac{d(x_2, x_2^{(n)})}{\beta}} \right)^\alpha \alpha \left[ \frac{d(x_2, x_2^{(n)}) - d(x_2 + \delta, x_2^{(n)})}{\beta + d(x_2, x_2^{(n)})} \right]$$

$$\text{Assuming } |x_2 - x_2^{(n)}| \gg \delta \quad \forall n \in [N] \tag{8}$$

$$\approx \left( \frac{1}{1 + \frac{d(x_2, x_2^{(n)})}{\beta}} \right)^\alpha \alpha \left[ \frac{-2\delta(x_2 - x_2^{(n)})}{\beta + d(x_2, x_2^{(n)})} \right] \tag{9}$$

Similarly,

$$\frac{\frac{\alpha}{\beta}(x_2 + \delta - x_2^{(n)})}{(1 + \frac{d(x_2+\delta, x_2^{(n)})}{\beta})^{\alpha+1}} - \frac{\frac{\alpha}{\beta}(x_2 - x_2^{(n)})}{(1 + \frac{d(x_2, x_2^{(n)})}{\beta})^{\alpha+1}}$$

$$\geq \frac{\alpha}{\beta}(x_2 - x_2^{(n)}) \left( \frac{1}{1 + \frac{d(x_2, x_2^{(n)})}{\beta}} \right)^{\alpha+1} (\alpha + 1) \left[ \frac{-2\delta(x_2 - x_2^{(n)})}{\beta + d(x_2, x_2^{(n)})} \right] + \frac{\delta \frac{\alpha}{\beta}}{(1 + \frac{d(x_2+\delta, x_2^{(n)})}{\beta})^{\alpha+1}}$$

$$\geq \frac{\alpha}{\beta}(x_2 - x_2^{(n)}) \left( \frac{1}{1 + \frac{d(x_2, x_2^{(n)})}{\beta}} \right)^{\alpha+1} (\alpha + 1) \left[ \frac{-2\delta(x_2 - x_2^{(n)})}{\beta + d(x_2, x_2^{(n)})} \right]$$

$$+ \frac{\delta \frac{\alpha}{\beta}}{(1 + \frac{d(x_2, x_2^{(n)})}{\beta})^{\alpha+1}} (\alpha + 1) \left[ \frac{-2\delta(x_2 - x_2^{(n)})}{\beta + d(x_2, x_2^{(n)})} + 1 \right]$$

$$= \frac{\alpha + 1}{(1 + \frac{d(x_2, x_2^{(n)})}{\beta})^{\alpha+1}} \left[ \frac{-2\delta(x_2 - x_2^{(n)})^2 \alpha/\beta - 2\delta^2 \alpha/\beta(x_2 - x_2^{(n)})}{\beta + d(x_2, x_2^{(n)})} + \frac{\delta \alpha}{\beta} \right]$$

$$= \frac{-2\delta\alpha(\alpha + 1)}{\beta(1 + \frac{d(x_2, x_2^{(n)})}{\beta})^{\alpha+1}} \left[ \frac{-2(x_2 - x_2^{(n)})[x_2 + \delta - x_2^{(n)}]}{\beta + d(x_2, x_2^{(n)})} + 1 \right] \tag{10}$$

Using inequalities 9, 10 in Equation 7, we have the following.

$$f(\mathbf{x} + \Delta\mathbf{x}) - f(\mathbf{x}) \geq \sum_n \left( \frac{1}{1 + \frac{d(x_1, x_1^{(n)})}{\beta}} \right)^\alpha \left( \frac{1}{1 + \frac{d(x_2, x_2^{(n)})}{\beta}} \right)^\alpha$$

$$\left[ \frac{-2\delta\alpha \tilde{y}_n}{\beta + d(x_2, x_2^{(n)})} + \frac{-2\delta\alpha(\alpha + 1)\tilde{y}_{n+N}}{\beta + d(x_2, x_2^{(n)})} \left( \frac{-2(x_2 - x_2^{(n)})[x_2 + \delta - x_2^{(n)}]}{\beta + d(x_2, x_2^{(n)})} + 1 \right) \right]$$

$$f(\mathbf{x} + \Delta\mathbf{x}) - f(\mathbf{x}) \geq \frac{2\delta\alpha}{\beta} \sum_n \left(\frac{1}{1 + \frac{d(x_1, x_1^{(n)})}{\beta}}\right)^\alpha \left(\frac{1}{1 + \frac{d(x_2, x_2^{(n)})}{\beta}}\right)^{\alpha+1}$$
$$\left[(\alpha + 1)\tilde{y}_{n+N} \left(\frac{2(x_2 - x_2^{(n)})[x_2 + \delta - x_2^{(n)}]}{\beta + d(x_2, x_2^{(n)})} - 1\right) - \tilde{y}_n\right] \quad (11)$$

Using the inequality $(1 + x)^r \geq 1 + rx$ if $r \leq 0$, we have

$$f(\mathbf{x} + \Delta\mathbf{x}) - f(\mathbf{x}) \geq \frac{2\delta\alpha}{\beta} \sum_n \left\{\left(1 - \frac{\alpha}{\beta}d(x_1, x_1^{(n)})\right)\left(1 - \frac{\alpha+1}{\beta}d(x_2, x_2^{(n)})\right)\right.$$
$$\left.\left[\frac{\alpha+1}{\beta}\tilde{y}_{n+N}\left(2(x_2 - x_2^{(n)})[x_2 + \delta - x_2^{(n)}](1 - d(x_2, x_2^{(n)})) - 1\right) - \tilde{y}_n\right]\right\}$$
$$= \frac{2\delta\alpha}{\beta}\Theta(x_1^2 x_2^6 + \delta x_1^2 x_2^5)$$

$\square$

# B   Proof of Theorem 2

We restate the result of Theorem 2 for clarity.

When we use an adversarial robustness algorithm to regularize the network, the fitted function has the following property.

$$|f(\mathbf{x} + [0, \delta]^T) - f(\mathbf{x})| \leq \frac{\alpha}{\beta}\delta_{max}f_{max}C$$
$$\text{where } C = \max_{\mathbf{x} \in \mathcal{X}} \min_{\hat{\mathbf{x}} \in \hat{\mathcal{X}}} |\mathbf{x}_2 - \hat{\mathbf{x}}_2|$$

$\delta_{max}$ and $f_{max}$ are maximum value of $\Delta x_2$ and $f(\mathbf{x})$ in the input domain ($\mathcal{X}$) respectively. $\hat{\mathcal{X}}$ denotes the subset of inputs covered by the robustness method. C therefore captures the maximum gap in coverage of the robustness method.

*Proof.* We begin by estimating the Lipschitz constant of a GP with squared exponential kernel.

$$f(\mathbf{x}) = K_{xX}K_{XX}^{-1}y$$
$$\frac{\partial f(x)}{\partial x_2} = \frac{\partial K_{xX}K_{XX}^{-1}y}{\partial x_2} = \tilde{K}_{xX}K_{XX}^{-1}y$$
$$\text{where } [\tilde{K}_{xX}]_n = \frac{\partial}{\partial x_2}\exp(-\frac{((x_1 - x_1^{(n)})^2 + (x_2 - x_2^{(n)})^2)}{2\theta^2})$$
$$= -\frac{(x_2 - x_2^{(n)})}{\theta^2}[K_{xX}]_n$$
$$\implies \frac{\partial f(x)}{\partial x_2} = -[\sum_{n=1}^{N} \frac{(x_2 - x_2^{(n)})}{\theta^2}[K_{xX}]_n]K_{XX}^{-1}y$$

We denote with $\delta_{max}$ the maximum deviation of any input from the training points, i.e. we define $\delta_{max}$ as $\max_{\mathbf{x} \in \mathcal{X}} \min_{n \in [N]} |x_2 - x_2^{(n)}|$. Also, we denote by $f_{max}$ the maximum function value in the input domain, i.e. $f_{max} \triangleq \max_{\mathbf{x} \in \mathcal{X}} f(\mathbf{x})$. We can then bound the partial derivative wrt second dimension as follows.

$$\frac{\partial f(\mathbf{x})}{\partial x_2} \leq \frac{\delta_{max}f(\mathbf{x})}{\theta^2} \leq \frac{\delta_{max}f_{max}}{\theta^2}$$

For any arbitrary point $\mathbf{x}$, the maximum function deviation is upper bounded by the product of maximum slope and maximum distance from the closest point covered by the adversarial distance method.

$$|f([x_1, x_2]^T) - f([x_1, \hat{x}_2]^T)| \leq \frac{\delta_{max}f_{max}}{\theta^2}\max_{\mathbf{x} \in \mathcal{X}} \min_{\hat{\mathbf{x}} \in \bar{\mathcal{X}}}|x_2 - \hat{x}_2| = \frac{\delta_{max}f_{max}}{\theta^2}C$$

Therefore,

$$|f(\mathbf{x} + [0, \delta]^T) - f(\mathbf{x})| \le 2 \frac{\delta_{max} f_{max}}{\theta^2} C$$

Marginalising $\theta^{-2}$ with the Gamma prior leads to the final form below.

$$|f(\mathbf{x} + [0, \delta]^T) - f(\mathbf{x})| \le 2C \frac{\alpha}{\beta} \delta_{max} f_{max}$$

$\square$

## C   Proof of Proposition 1

We restate the result here for clarity.

*Consider a regression task with $D + 1$-dimensional inputs $\mathbf{x}$ where the first $D$ dimensions are irrelevant, and assume they are $x_d = y, d \in [1, D]$ while $x_{D+1} \sim \mathcal{N}(y, 1/K)$. The MAP estimate of linear regression parameters $f(\mathbf{x}) = \sum_{d=1}^{D+1} w_d x_d$ when fitted using Avg-Ex are as follows: $w_d = 1/(D + K), \quad d \in [1, D]$ and $w_{D+1} = K/(K + D)$.*

*Proof.* Without loss of generality, we assume $\alpha, \sigma^2$ parameters of Avg-Ex are set to 1. In effect, our objective is to fit parameters that predict well for inputs sampled using standard normal perturbations, i.e. $\mathbf{x}^{(n)} + \mathbf{m}\epsilon, \forall n \in [1, N], \epsilon \sim \mathcal{N}(0, 1), \mathbf{m} = [1, 1, \ldots, 1, 0]^T \in \{0, 1\}^{D+1}$. The original problem therefore is equivalent to fitting on transformed input $\hat{\mathbf{x}}$ such that $\hat{\mathbf{x}}_i^{(n)} \sim \mathcal{N}(y, \sigma_i^2)$ where $\sigma_i^2 = 1$ for all $i \le D$ and is $1/K$ when $i = D + 1$.

Likelihood of observations for the equivalent problem is obtained as follows.

$$
\begin{aligned}
P(y \mid \hat{x}_1, \hat{x}2, \ldots, \hat{x}_{D+1}) &= \prod_{i=1}^{D+1} P(y \mid \hat{x}_i) \propto \prod_{i=1}^{D+1} P(\hat{x}_i \mid y) P(y) \\
&= \prod_i \mathcal{N}(\hat{x}_i; y, \sigma_i^2) \propto \exp\left(-\sum_i \frac{(y - \hat{x}_i)^2}{2\sigma_i^2}\right) \\
&= \exp\left\{ -y^2 \left(\sum_i \frac{1}{2\sigma_i^2}\right) + y\left(\sum_i \frac{\hat{x}_i}{\sigma_i^2}\right) + \sum_i \frac{\hat{x}_i^2}{2\sigma_i^2} \right\} \\
&\propto \mathcal{N}\left(y; \sum_i \frac{\hat{x}_i}{\sigma_i^2} P, P\right) \\
\text{where } P &= \frac{1}{\sum_i 1/\sigma_i^2}
\end{aligned}
$$

Substituting, the value of $\sigma_i$ defined as above, we have P=D+K and the MLE estimate for the linear regression parameters are as shown in the statement. The MAP estimate also remains the same since we do not impose any informative prior on the regression weights. $\square$

## D   Parametric Model Analysis

In this section we show that a similar result to what is shown for non-parametric models also holds for parametric models. We will analyse the results for a two-layer neural networks with ReLU activations. We consider a more general case of $D$ dimensional input where the first $d$ dimensions identify the spurious features. We wish to fit a function $f : \mathbb{R}^D \to \mathbb{R}$ such that $f(\mathbf{x})$ is robust to perturbations to the spurious features. We have the following bound when training a model using gradient regularization of Ross et al. (2017).

**Proposition 2.** *We assume that the model is parameterised as a two-layer network with ReLU activations such that $f(\mathbf{x}) = \sum_j \beta_j \phi(\sum_i w_{ji} x_i + b_j)$ where $\vec{\beta} \in \mathbb{R}^F, \vec{w} \in \mathbb{R}^{F \times D}, \vec{b} \in \mathbb{R}^F$ are the parameters, and $\phi(z) = \max(z, 0)$ is the ReLU activation. For any function such that gradients*

*wrt to the first d features is exactly zero, i.e. $\frac{\partial f}{\partial x_i}|_{\mathbf{x}_i^{(n)}} = 0 \quad \forall i \in [1, d], n \in [1, N]$, we have the following bound on the function value deviations for input perturbations from a training instance $\mathbf{x}$: $\tilde{x} - x = \Delta\mathbf{x} = [\Delta\mathbf{x}_{1:d}^T, \mathbf{0}_{d+1:D}^T]^T$.*

$$|f(\tilde{x}) - f(x)| = \Theta((\|\vec{\beta}\|^2 + \|\vec{w}\|_F^2)\|\Delta\mathbf{x}\|) \tag{12}$$

For a two-layer network trained to regularize gradients wrt first d dimensions on training data, the function value deviation from an arbitrary point $\tilde{\mathbf{x}}$ from a training point $\mathbf{x}$ such that $\tilde{\mathbf{x}} - \mathbf{x} = \Delta\mathbf{x} = [\Delta\mathbf{x}_{1:d}^T, \mathbf{0}_{d+1:D}^T]^T$ is bounded as follows.

$$|f(\tilde{x}) - f(x)| = \Theta((\|\vec{\beta}\|^2 + \|\vec{w}\|_F^2)\|\Delta\mathbf{x}\|)$$

*Proof.* Recall that the function is parameterised using parameters $\vec{w}, \vec{b}, \vec{\beta}$ such that $f(\mathbf{x}) = \sum_j \beta_j \phi(\sum_i w_{ji}x_i + b_j)$ where $\vec{\beta} \in \mathbb{R}^F, \vec{w} \in \mathbb{R}^{F \times D}, \vec{b} \in \mathbb{R}^F$ are the parameters, and $\phi(z) = \max(z, 0)$ is the ReLU activation.

Since we train such that $\frac{\partial f(\mathbf{x})}{\partial x_i} = 0, \quad i \in [1, d]$, we have that $\frac{\partial f(\mathbf{x})}{x_i} = \sum_j \beta_j \hat{\phi}(\sum_i w_{ij}x_i + b_i)w_{ij}$ where $\hat{\phi}(a) = \max(\frac{a}{|a|}, 0)$.

We now bound the variation in the function value for changes in the input when moving from $\mathbf{x} \to \tilde{\mathbf{x}}$ where $\mathbf{x}$ is an instance from the training data. We define four groups of neurons based on the sign of $\sum_i w_{ji}x_i + b_j$ and $\sum_i w_{ji}\tilde{x}_i + b_j$. $g_1$ is both positive, $g_2$ is negative and positive, $g_3$ is positive and negative, $g_4$ is both negative. By defining groups, we can omit the ReLU activations as below.

$$f(\tilde{\mathbf{x}}) - f(\mathbf{x}) = \sum_j \beta_j \phi(\sum_i w_{ji}\tilde{x}_i + b_j) - \sum_j \beta_j \phi(\sum_i w_{ji}x_i + b_j)$$

$$= \sum_{j \in g_1} \beta_j \sum_i w_{ji}(\tilde{x}_i - x_i) + \sum_{j \in g_2} \beta_j(\sum_i w_{ji}\tilde{x}_i + b_j) - \sum_{j \in g_3} \beta_j(\sum_i w_{ji}x_i + b_j)$$

$$= \sum_{j \in g_1} \beta_j \sum_{i=1}^d w_{ji}(\tilde{x}_i - x_i) + \sum_{j \in g_2} \beta_j(\sum_{i=1}^D w_{ji}\tilde{x}_i + b_j) - \sum_{j \in g_3} \beta_j(\sum_{i=1}^D w_{ji}x_i + b_j)$$

Since we have that $\sum_{j \in g_1 \cup g_3} \beta_j w_{ij} = 0, \forall i \in [1, d]$, we have

$$= \sum_{j \in g_1} \beta_j \sum_{i=1}^d w_{ji}\tilde{x}_i + \sum_{j \in g_2} \beta_j(\sum_{i=1}^d w_{ji}\tilde{x}_i + \sum_{i=d+1}^D w_{ji}x_i + b_j) - \sum_{j \in g_3} \beta_j(\sum_{i=d+1}^D w_{ji}x_i + b_j)$$

$$\underbrace{- \sum_{j \in g_1} \beta_j \sum_{i=1}^d w_{ji}x_i - \sum_{j \in g_3} \beta_j \sum_{i=1}^d w_{ji}x_i}_{=\sum_{i=1}^d x_i \sum_{j \in g_1 \cup g_3} \beta_j w_{ji} = 0}$$

$$= \sum_{j \in g_1 \cup g_2} \beta_j \sum_{i=1}^d w_{ji}\tilde{x}_i + \sum_{j \in g_2} \beta_j(\sum_{i=d+1}^D w_{ji}x_i + b_j) - \sum_{j \in g_3} \beta_j(\sum_{i=d+1}^D w_{ji}x_i + b_j)$$

retaining only the terms that depend on $\Delta x = \tilde{x} - x$, the expression is further simplified as a term that grows with $\Delta\mathbf{x}$ and a constant term that depends on the value of $\mathbf{x}$

$$= \sum_{j \in g_1 \cup g_2} \beta_j \sum_{i=1}^d w_{ji}\Delta x_i + \text{constant}$$

$$\implies = \Theta(\|\beta\|\|\vec{w}\|_F\|\Delta\mathbf{x}\|) \quad \text{Cauchy-Schwartz inequality}$$

$$= \Theta((\|\beta\|^2 + \|\vec{w}\|_F^2\|)\|\Delta\mathbf{x}\|)$$

$\square$

# E    Further Experiment Details

## E.1    Hyperparameters.

We picked the learning rate, optimizer, weight decay, and initialization for best performance with ERM baseline on validation data, which are not further tuned for other baselines unless stated otherwise. We picked the best $\lambda$ for Grad-Reg and CDEP from [1, 10, 100, 1000]. Additionally, we also tuned $\beta$ (weight decay) for Grad-Reg from [1e-4, 1e-2, 1, 10]. For Avg-Ex, perturbations were drawn from 0 mean and $\sigma^2$ variance Gaussian noise, where $\sigma$ was chosen from [0.03, 0.3, 1, 1.5, 2]. In PGD-Ex, the worst perturbation was optimized from $\ell_\infty$ norm $\epsilon$-ball through seven PGD iterations, where the best $\epsilon$ is picked from the range 0.03-5. We did not see much gains when increasing PGD iterations beyond 7, Appendix F contains some results when the number of iterations is varied. In IBP-Ex, we follow the standard procedure of Gowal et al. (2018) to linearly dampen the value of $\alpha$ from 1 to 0.5 and linearly increase the value of $\epsilon$ from 0 to $\epsilon_{max}$, where $\epsilon_{max}$ is picked from 0.01 to 2. We usually just picked the maximum possible value for $\epsilon_{max}$ that converges. For IBP-Ex+Grad-Reg, we have the additional hyperparameter $\lambda$ (Eqn. 4), which we found to be relatively stable and we set it to 1 for all experiments.

## E.2    Metrics

**Relative Core Sensitivity (RCS) (Singla et al., 2022)**. The metric measures the relative dependence of the model on core features and is normalised such that the best value is 100. Higher value of RCS imply that the model is exploiting core features more than the spurious features.

$$RCS = 100 \times \frac{acc^{(C)} - acc^{(S)}}{2\min(\bar{a}, 1 - \bar{a})}, \text{ where } \bar{a} = \frac{acc^{(C)} + acc^{(S)}}{2}$$

$$acc^C \triangleq \frac{1}{N}\sum_n \mathbb{1}\left(f\left(\mathbf{x}^{(n)} + \sigma(\mathbf{z} \odot \mathbf{m}^{(n)}); \theta\right) = y^{(n)}\right)$$

$$acc^S \triangleq \frac{1}{N}\sum_n \mathbb{1}\left(f\left(\mathbf{x}^{(n)} + \sigma(\mathbf{z} \odot (1 - \mathbf{m}^{(n)})); \theta\right) = y^{(n)}\right)$$

where $\mathbf{z} \sim \mathcal{N}(\mathbf{0}, \mathbf{I})$ and $\sigma = 0.25$ for all experiments. The interpretation of $acc^{(C)}$ is the accuracy when noise is added outside of core region, and $acc^{(S)}$ is the accuracy when noise is added outside of spurious region.

## E.3    Data splits

We randomly split available labelled data in to training, validation, and test sets in the ratio of (0.75, 0.1, 0.15) for ISIC, (0.65, 0.1, 0.25) for Plant (similar to Schramowski et al. (2020)) and (0.6, 0.15, 0.25) for Salient-Imagenet. We use the standard train-test splits on MNIST.

## E.4    Datasets

**ISIC dataset** The ISIC dataset consists of 2,282 cancerous (C) and 19,372 non-cancerous (NC) skin cancer images of 299 by 299 size, each with a ground-truth diagnostic label. We follow the standard setup and dataset released by Rieger et al. (2020), which included masks with patch segmentations. In half of the NC images, there is a spurious correlation in which colorful patches are only attached next to the lesion. This group is referred to as patch non-cancerous (PNC) and the other half is referred to as not-patched non-cancerous (NPNC) Codella et al. (2019). Since trained models tend to learn easy-to-learn and useful features, they tend to take a shortcut by learning spurious features instead of understanding the desired diagnostic phenomena. Therefore, our goal is to make the model invariant to such colorful patches by providing a human specification mask indicating where they are.

**decoy-MNIST dataset** The MNIST dataset consists of 70,000 images of handwriting digit from 0 to 9. Each class has about 7,000 images of 28 by 28 size. We use three-fully connected layers for multi classification with 512 hidden dimension and 3 channels.

**Salient-Imagenet.** The six classes we considered are *Rhinoceros Beetle, Dowitcher, Alaskan Tundra Wolf, Dragonfly, Gorilla, Snoek Fish*.

## E.5   Computational cost

**Run time and memory usage** Table 3 presents the computation costs, including run time and memory usage, for each method using GTX 1080 Ti. It is worth noting that IBP-Ex has significantly less run time and memory usage compared to PGD-Ex, with a 10-fold reduction in run time and a 2.5-fold reduction in memory usage. Considering that PGD-Ex and IBP-Ex have similar performance in terms of worst group accuracy, as shown in Table 5, IBP-Ex+Grad-Reg appears to be comparably effective and efficient for model modification. Additionally, the combined method IBP-Ex+Grad-Reg, which presents the best performance in terms of averaged and worst group accuracy compared to PGD-Ex, also has a 3-fold reduction in run time and a 2-fold reduction in memory usage compared to PGD-Ex.

| Grad-Reg | PGD-Ex | IBP-Ex | IBP-Ex+Grad-Reg | PGD-Ex+Grad-Reg |
|---|---|---|---|---|
| ×2.3 | ×4.9 | ×2.2 | ×3.5 | × 7.0 |

Table 3: Running time in comparison to ERM on the ISIC dataset

## E.6   Network Architecture

**Model architecture on the decoy-MNIST dataset**

```
Sequential(
    (0): Conv2d(3, 32, kernel_size=(3, 3), stride=(1, 1), padding=(1, 1))
    (1): ReLU()
    (2): Conv2d(32, 32, kernel_size=(4, 4), stride=(2, 2), padding=(1, 1))
    (3): ReLU()
    (4): Conv2d(32, 64, kernel_size=(3, 3), stride=(1, 1), padding=(1, 1))
    (5): ReLU()
    (6): Conv2d(64, 64, kernel_size=(4, 4), stride=(2, 2), padding=(1, 1))
    (7): ReLU()
    (8): Flatten(start_dim=1, end_dim=-1)
    (9): Linear(in_features=200704, out_features=1024, bias=True)
    (10): ReLU()
    (11): Linear(in_features=1024, out_features=1024, bias=True)
    (12): ReLU()
    (13): Linear(in_features=1024, out_features=2, bias=True)
  )
```

**Model architecture on the ISIC dataset**

```
Sequential(
    (0): Flatten(start_dim=1, end_dim=-1)
    (1): Linear(in_features=2352, out_features=512, bias=True)
    (2): ReLU()
    (3): Linear(in_features=512, out_features=512, bias=True)
    (4): ReLU()
    (5): Linear(in_features=512, out_features=512, bias=True)
    (6): ReLU()
    (7): Linear(in_features=512, out_features=10, bias=True)
    )
```

**Model architecture on the Plant phenotyping dataset**

```
Sequential(
    (0): Conv2d(3, 32, kernel_size=(3, 3), stride=(1, 1), padding=(1, 1))
    (1): ReLU()
    (2): Conv2d(32, 32, kernel_size=(4, 4), stride=(2, 2), padding=(1, 1))
    (3): ReLU()
    (4): Conv2d(32, 64, kernel_size=(3, 3), stride=(1, 1), padding=(1, 1))
    (5): ReLU()
    (6): Conv2d(64, 64, kernel_size=(4, 4), stride=(2, 2), padding=(1, 1))
```

```
(7): ReLU()
(8): Flatten(start_dim=1, end_dim=-1)
(9): Linear(in_features=200704, out_features=1024, bias=True)
(10): ReLU()
(11): Linear(in_features=1024, out_features=1024, bias=True)
(12): ReLU()
(13): Linear(in_features=1024, out_features=2, bias=True)
)
```

# F    Additional Results

## F.1    Standard deviations

We repeated all our experiments on Decoy-MNIST, Plant and ISIC dataset three times and report the mean and standard deviation in Table 4. Similarly, we report in Table 5, the standard deviations for the corresponding Table 2 of the main paper.

| Dataset→ | Decoy-MNIST | | Plant | | ISIC | |
|---|---|---|---|---|---|---|
| Method↓ | Avg Acc | Wg Acc | Avg Acc | Wg Acc | Avg Acc | Wg Acc |
| ERM | $15.1 \pm 1.3$ | $10.5 \pm 5.4$ | $71.3 \pm 2.5$ | $54.8 \pm 1.3$ | $77.3 \pm 2.4$ | $55.9 \pm 2.3$ |
| G-DRO | $64.1 \pm 0.1$ | $28.1 \pm 0.1$ | $74.2 \pm 5.8$ | $58.0 \pm 4.6$ | $66.6 \pm 5.4$ | $58.5 \pm 10.7$ |
| Grad-Reg | $72.5 \pm 1.7$ | $46.2 \pm 1.1$ | $72.4 \pm 1.3$ | $68.2 \pm 1.4$ | $76.4 \pm 2.4$ | $60.2 \pm 7.4$ |
| CDEP | $14.5 \pm 1.8$ | $10.0 \pm 0.7$ | $67.9 \pm 10.3$ | $54.2 \pm 24.7$ | $73.4 \pm 1.0$ | $60.9 \pm 3.0$ |
| Avg-Ex | $29.5 \pm 0.3$ | $19.5 \pm 1.4$ | $76.3 \pm 0.3$ | $64.5 \pm 0.3$ | $77.1 \pm 2.1$ | $55.2 \pm 6.6$ |
| PGD-Ex | $67.6 \pm 1.6$ | $51.4 \pm 0.3$ | $79.8 \pm 0.3$ | $78.5 \pm 0.3$ | $\mathbf{78.7 \pm 0.5}$ | $\mathbf{64.4 \pm 4.3}$ |
| IBP-Ex | $68.1 \pm 2.2$ | $47.6 \pm 2.0$ | $76.6 \pm 3.5$ | $73.8 \pm 1.7$ | $75.1 \pm 1.2$ | $64.2 \pm 1.2$ |
| P+G | $\mathbf{96.9 \pm 0.3}$ | $\mathbf{95.8 \pm 0.4}$ | $79.4 \pm 0.5$ | $76.7 \pm 2.8$ | $\mathbf{79.6 \pm 0.5}$ | $\mathbf{67.5 \pm 1.1}$ |
| I+G | $\mathbf{96.9 \pm 0.2}$ | $95.0 \pm 0.6$ | $\mathbf{81.7 \pm 0.2}$ | $\mathbf{80.1 \pm 0.3}$ | $78.4 \pm 0.5$ | $65.2 \pm 1.8$ |

Table 4: Macro-averaged (Avg) accuracy and worst group (Wg) accuracy on (a) decoy-MNIST, (b) plant dataset, (c) ISIC dataset. Results are averaged over three runs and their standard deviation is shown after $\pm$. I+G is short for IBP-Ex+Grad-Reg and P+G for PGD-Ex+Grad-Reg. See text for more details.

| Method | NPNC | PNC | C | Avg | Wg |
|---|---|---|---|---|---|
| ERM | $55.9 \pm 2.3$ | $96.5 \pm 2.4$ | $79.6 \pm 6.6$ | $77.3 \pm 2.4$ | $55.9 \pm 2.3$ |
| G-DRO | $72.4 \pm 4.0$ | $63.2 \pm 14.8$ | $64.1 \pm 5.6$ | $66.6 \pm 5.4$ | $58.5 \pm 10.7$ |
| Grad-Reg | $67.1 \pm 4.8$ | $99.0 \pm 1.0$ | $63.2 \pm 11.3$ | $76.4 \pm 2.4$ | $60.2 \pm 7.4$ |
| CDEP | $72.1 \pm 5.4$ | $98.9 \pm 0.7$ | $62.2 \pm 4.7$ | $73.4 \pm 1.0$ | $60.9 \pm 3.0$ |
| Avg-Ex | $62.3 \pm 11.7$ | $97.8 \pm 0.8$ | $71.0 \pm 16.7$ | $77.1 \pm 2.1$ | $55.2 \pm 6.6$ |
| PGD-Ex | $65.4 \pm 5.4$ | $99.0 \pm 0.3$ | $71.7 \pm 6.7$ | $\mathbf{78.7 \pm 0.5}$ | $64.4 \pm 4.3$ |
| IBP-Ex | $68.4 \pm 3.4$ | $98.5 \pm 1.0$ | $67.7 \pm 4.8$ | $75.1 \pm 1.2$ | $64.2 \pm 1.2$ |
| P+G | $69.6 \pm 2.8$ | $98.84 \pm 0.6$ | $70.4 \pm 4.1$ | $\mathbf{79.6 \pm 0.5}$ | $\mathbf{67.5 \pm 1.1}$ |
| I+G | $66.6 \pm 3.1$ | $99.6 \pm 0.2$ | $68.9 \pm 4.7$ | $\mathbf{78.4 \pm 0.5}$ | $65.2 \pm 1.8$ |

Table 5: Macro-averaged (Avg) accuracy and worst group (Wg) accuracy on ISIC dataset. Also shown are the average precision scores for each of the three groups. All the results are averaged over three runs and their standard deviation is shown after $\pm$. Note that the worst group for each run can be different

## F.2    Additional results on Salient-Imagenet

In Table 6, we show average accuracy and accuracy when noise (drawn from standard normal) is added to spurious or irrelevant regions (N-Acc column). We observe that for PGD-Ex + Grad-Reg, the accuracy did not diminish by much when noise is added to the spurious region.

## F.3    Comparison of PGD-Ex and IBP-Ex

In Table 5, it is difficult to compare the worst group accuracy of IBP-Ex (64.2) and PGD-Ex (64.4) due to the comparably high standard deviation of PGD-Ex (4.3). Therefore, we additionally compare

| Method | Accuracy | N-Acc ↑ | RCS ↑ |
|---|---|---|---|
| ERM | 96.4 | 87.5 | 47.9 |
| Grad-Reg | 88.3 | 82.2 | 52.5 |
| PGD-Ex | 93.8 | 90.2 | 58.7 |
| PGD-Ex + Grad-Reg | **94.6** | **93.8** | **65.0** |

Table 6: The columns in that order are the average accuracy, accuracy when noise is added to spurious (or irrelevant) regions and RCS value for our Salient-Imagenet data setup.

the accuracy drop when colorful patches are removed from images in the PNC group in Table 7. We replace the colorful patch of the image with its mean value, making it looks like a background skin color. Note that we evaluate the robustness to concept-level perturbations rather than pixel-level perturbations, as our focus is on avoiding spurious concept features rather than robustness to adversarial attacks. Interestingly, the accuracy drops about 17% and 37% in IBP-Ex and PGD-Ex, respectively, showing that IBP-Ex is more robust to concept perturbations. This can be explained by the effectiveness of robustness methods in covering the epsilon ball with the center of each input point defined in a low-dimensional manifold annotated in the human specification mask. IBP guarantees robustness on any possible pixel combination within the epsilon ball while PGD only considers the worst case in the epsilon ball. When the inner maximization to find the PGD attack is non-convex, an inappropriate local worst case is found instead of the global one. Thus, IBP-Ex shows better robustness when spurious concepts are removed, which involves large perturbations on irrelevant parts within the defined epsilon ball. The combined method IBP-Ex+Grad-Reg, where Grad-Reg compensates for the practical limitations of the training procedure of IBP-Ex, shows about 1% higher worst group accuracy than IBP-Ex alone.

| Method | PNC | PNC (Remove patch) |
|---|---|---|
| PGD-Ex | $99.0 \pm 0.3$ | $62.2 \pm 17.0$ |
| IBP-Ex | $98.5 \pm 1.0$ | $81.6 \pm 16.5$ |
| IBP-Ex+Grad-Reg | $99.6 \pm 0.2$ | **$82.5 \pm 9.5$** |

Table 7: Comparison between robustness based methods. Macro-averaged accuracy and regval loss before and after removing color patch part of images in PNC group on ISIC dataset.

### F.4 Results of PGD-Ex with different epsilon and iteration number.

We experimented with different values of epsilon and iteration numbers on the ISIC and Plant phenotyping datasets. The epsilon values tested were 0.03, 0.3, 1, 3, and 5, and the iteration numbers were 7 and 25. In Figure 4, the results on the ISIC dataset showed that using an iteration of 7 with different epsilon values resulted in stable results, but using an iteration of 25 resulted in unstable worst group accuracy. However, in the Plant phenotyping dataset, we found that both average and worst group accuracy were similar regardless of the epsilon and iteration values used.

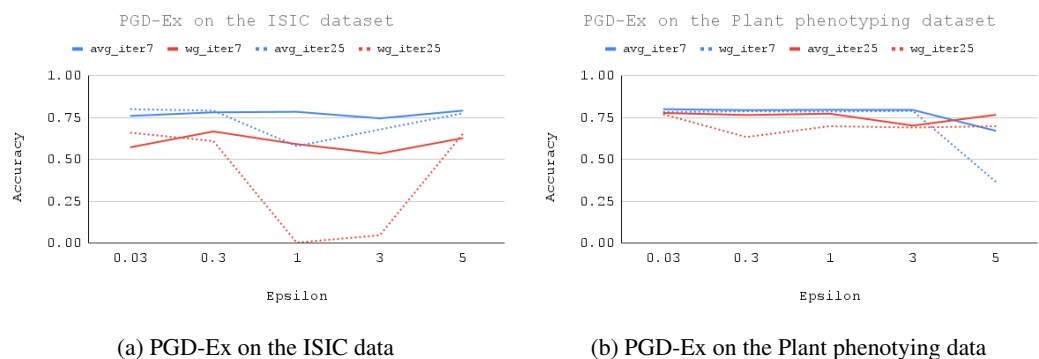

(a) PGD-Ex on the ISIC data          (b) PGD-Ex on the Plant phenotying data

Figure 4: PGD-Ex results on the ISIC and Plant phenotyping dataset with different epsilon and iteration numbers in (a) and (b), respectively.

### F.5 Out-of-distribution scenarios on the Plant data

In the main paper, we follow the dataset construction from Schramowski et al. (2020) to replace background with the average pixel value, which is obtained from train split. Here, in Table 8 we evaluated on a test set obtained by adding varying magnitude of noise to the background to test methods under out-of-distribution scenarios. We observe that robustness and regularization methods when combined led to a model that is far more robust to noise in the background, aligning with our original results on the plant dataset.

| | Noise ($\mathcal{N}(0, 1)$) | | Noise ($\mathcal{N}(0, 10)$) | | Noise ($\mathcal{N}(0, 30)$) | |
|---|---|---|---|---|---|---|
| | Avg Acc | Wg Acc | Avg Acc | Wg Acc | Avg Acc | Wg Acc |
| ERM | 59.8 ± 11.9 | 43.5 ± 2.0 | 57.4 ± 7.6 | 38.1 ± 4.8 | 55.8 ± 1.7 | 22.0 ± 3.7 |
| Grad-Reg | 71.6 ± 2.0 | 66.1 ± 1.8 | 68.7 ± 6.2 | 53.4 ± 4.3 | 56.1 ± 3.3 | 34.8 ± 1.8 |
| PGD-Ex+Grad-Reg | 69.8 ± 1.8 | 67.2 ± 2.1 | 69.5 ± 3.7 | 60.6 ± 4.8 | 67.5 ± 4.5 | 50.8 ± 2.4 |

Table 8: Out-of-distribution scenarios on the Plant data

### F.6 Generality to new explanation methods: Integrated-gradient and CDEP

In Table 9, we introduced evaluation using Integrated-gradient (Sundararajan et al., 2017) based regularization and also added evaluation with PGD-Ex+CDEP that we did not originally include on Decoy-MNIST dataset.

| Alg. | Avg Acc | Wg Acc |
|---|---|---|
| Integrated-Grad | 26.7 ±1.3 | 17.6 ±1.2 |
| CDEP | 14.5 ±1.8 | 10.0 ±0.7 |
| PGD-Ex | 67.6 ±1.6 | 51.4 ±0.3 |
| PGD-Ex+Integrated-Grad | 80.5 ±2.1 | **62.1 ± 6.8** |
| PGD-Ex+CDEP | **84.8 ±0.8** | **64.2 ±1.6** |

Table 9: PGD-Ex+Integrated-Grad and PGD-Ex+CDEP on the Decoy-MNIST data

### F.7 Generality to new model architecture: Attention map-based (ViT)

Using a Visual transformer architecture (of depth 3 and width 128), we evaluated regularization using local explanations obtained using an attention map – regularization based on attention maps was used to supervise prior knowledge in Miao et al. (2022) and is called SPAN. We obtained saliency explanations on inputs using the procedure proposed in Miao et al. (2022), shown in Table 10

| Alg. | Avg Acc | Wg Acc |
|---|---|---|
| ERM | 10.0 ±0.3 | 8.1 ±0.3 |
| SPAN | 19.0 ±0.3 | 8.1 ±0.3 |
| PGD-Ex | **64.6 ±4.7** | **37.4 ±3.5** |
| PGD-Ex+SPAN | **63.1 ±2.6** | **39.4 ± 2.9** |

Table 10: Attention Map based local explanations with ViT on the Decoy-MNIST data

### F.8 Sensitivity to hyperparameters on Decoy-MNIST dataset

In Table 11, we show sensitivity to hyperparameters on Decoy-MNIST dataset. In summary, results are broadly stable with the choice of hyperparameters, and we did not extensively search for the best hyperparameters.

| Decoy-MNIST | Lambda (Grad-Reg) | Eps (PGD-Ex) | Avg -Acc | Wg -Acc |
|---|---|---|---|---|
| PGD-Ex + Grad-Reg | **1** | **3** | **96.9 ± 0.3** | **95.8 ± 0.4** |
| | 0.1 | 3 | 96.8 ± 0.8 | 94.2 ± 0.2 |
| | 1.5 | 3 | 95.6 ± 1.0 | 93.0 ± 1.0 |
| | 5 | 3 | 91.6 ± 0.9 | 87.6 ± 2.3 |
| | 0.0001 | 3 | 75.5 ± 0.9 | 57.2 ± 3.6 |
| | 1 | 1 | 95.1 ± 2.0 | 91.3 ± 3.9 |
| | 1 | 2 | 95.4 ± 1.8 | 92.0 ± 1.6 |
| | 1 | 4 | 97.5 ± 0.2 | 95.4 ± 0.8 |
| | 1 | 5 | 93.8 ± 1.6 | 86.3 ± 3.1 |
| | 1 | 0.1 | 58.5 ± 7.9 | 30.0 ± 2.7 |
| | 1 | 0.0001 | 59.5 ± 1.1 | 40.1 ± 2.0 |
| PGD-Ex | **0** | **3** | **67.6 ± 1.6** | **51.4 ± 0.3** |
| | 0 | 0.0001 | 16.5 ± 1.1 | 15.4 ± 2.7 |
| | 0 | 0.1 | 19.1 ± 0.7 | 13.6 ± 0.6 |
| | 0 | 1 | 62.5 ± 1.0 | 40.1 ± 2.2 |
| | 0 | 2 | 74.6 ± 5.6 | 52.8 ± 4.8 |
| | 0 | 4 | 71.9 ± 8.5 | 58.6 ± 12.9 |
| | 0 | 5 | 57.0 ± 3.1 | 42.6 ± 4.2 |
| Grad-Reg | **10** | **0** | **64.1 ± 0.1** | **28.1 ± 0.1** |
| | 5 | 0 | 39.2 ± 2.2 | 21.5 ± 0.6 |
| | 20 | 0 | 49.1 ± 2.7 | 33.2 ± 4.3 |
| | 100 | 0 | 50.1 ± 0.4 | 35.2 ± 2.3 |
| | 500 | 0 | 48.0 ± 0.9 | 36.2 ± 1.7 |
| | 1000 | 0 | 49.6 ± 1.7 | 30.5 ± 6.1 |

Table 11: Sensitivity to hyperparameters on Decoy-MNIST dataset

## G   Discussion on poor CDEP performance

**Regarding ISIC dataset discrepancy:** In Table 5, CDEP demonstrates better performance in worst group accuracy compared to ERM on the ISIC dataset. However, it fails to surpass RRR, which contradicts results from previous research in Rieger et al. (2020) where CDEP was found to perform better than RRR. This discrepancy may be attributed to the fact that Rieger et al. (2020) used different metrics (F1 and AUC) and employed a pretrained VGG model to estimate the contribution of mask features, whereas in our study we used worst group accuracy and employed a four-layer CNN followed by three fully connected layers without any pretraining. We do not use a pre-trained model for CDEP in order to make a fair comparison to other methods. As a result, CDEP also fails to improve worst group accuracy over ERM on the Plant Phenotyping and Decoy-MNIST datasets. We further illustrate the interpretations of CDEP on the Plant Phenotyping dataset using Smooth Gradient in Figure 5. In comparison to the interpretations of other methods shown in Figure 3 in the main paper, CDEP appears to focus primarily on the spurious agar part instead of the main leaf part.

**Regarding DecoyMNIST dataset discrepancy:** Note that our Decoy-MNIST setting is inspired from decoy-mnist of CDEP (Rieger et al., 2020), but not the same. All the methods were found to be equally good on the original decoy-mnist dataset (Rieger et al., 2020), which is why we had to alter the dataset to be more challenging. A key difference is that the volume of spurious/simple features in our version of decoy-mnist dataset is much higher, making it harder to remove dependence of a model on decoy/spurious features. This explains why there is the performance gap on this dataset reported in our paper and CDEP (Rieger et al., 2020).

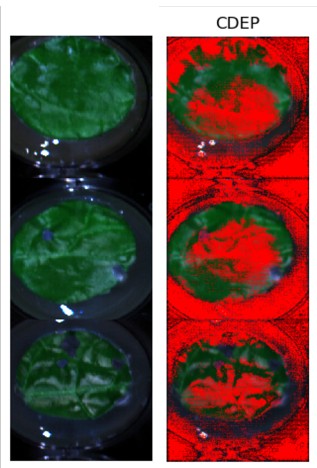

Figure 5: Visual heatmap of salient features for CDEP on three sample images from the train split of Plant phenotyping data. Importance score from SmoothGrad Smilkov et al. (2017) method is normalized between 0 to 1 and visualized with a threshold 0.6.

