# OpenReview forum: "Use perturbations when learning from explanations"
_NeurIPS.cc/2023/Conference — NeurIPS 2023 poster_

### Official Review · Reviewer_WA6d · 2023-06-23

**Soundness:** 4 excellent
**Presentation:** 3 good
**Contribution:** 3 good
**Rating:** 7
**Confidence:** 4

**Summary:**

This paper proposes a novel approach to Machine Learning from Explanations (MLX), the setting in which data points for learning are paired with human-annotated “explanations”, usually in the form of image masks marking regions that are known to be irrelevant with respect to the downstream prediction task. Previous methods for MLX made use of regularization-based techniques, forcing the learnt function to be locally independent of the irrelevant features for the training data points. The authors show that these methods require strong model smoothing to be effective, to the extent that downstream task performance is heavily penalized. They then propose “robustness-based” methods, which force the model to be robust to perturbation across the irrelevant features, and show theoretically and practically that these techniques work even without heavy regularization smoothing. The authors then show, using three benchmarks, that their technique works in practice, but best results are obtained by combining robustness and regularization-based methods.

**Strengths:**

The paper improves on methods from an existing problem setting, applies robustness techniques where they had not been tried before, and convincingly shows both with theory and experiments that such techniques work.

The theorems, despite their simplified setting, intuitively demonstrate why the approach works better than the alternatives.

The experimental results convincingly show that the robustness-based techniques work better than regularization-based techniques.

**Weaknesses:**

While the contribution itself is nicely self-contained and perfectly valid when taken at face value (that in the MLX setting, training GPs or simple CNNs, robustness-based methods are better than regularization-based methods), one cannot help but wonder whether the entire setup is relevant or interesting, considering current state of the art architectures and how they attend to their input. Attention-based architectures like ViTs, due to intermediate-layer attention maps, lend themselves to very natural investigations commensurable to the kind of pixel-wise explanations used in this work.

One can imagine that, as these transformer models also easily incorporate masking, the whole MLX problem could be reframed to be about either manipulating attention maps, or dynamically learning attention masks. And example is [Yang et al. (2023)](https://openaccess.thecvf.com/content/CVPR2023/html/Yang_Improving_Visual_Grounding_by_Encouraging_Consistent_Gradient-Based_Explanations_CVPR_2023_paper.html). Addressing these concerns either in the introduction or related work could help contextualize the paper’s contributions.

**Questions:**

It would be interesting, to place the present work in perspective, to at least spend some time in related work to investigate how “explanations” are currently treated in the literature on transformer-based image models like ViTs. Is there any existing literature showing techniques for MLX (maybe not labeled as such) that are dependent on a particular architecture choice? If so, what makes the present work still interesting?

Edit: Increased the score after author rebuttal addressing the points above - see comment below.

**Limitations:**

The authors briefly mention limitations related to the possibility, in future work, to learn from incomplete explanation data. It would be interesting to add a segment related to the points raised in Weaknesses and Questions.

---

> ### Author Rebuttal · Authors · 2023-08-09
>
> We thank the reviewer for their passionate assessment of our work. We are very glad that the reviewer is appreciative of the novelty of our contribution, intuition from our analysis and completeness of experiments. We enjoyed addressing their concerns on the future relevance of our contributions. We will make sure to include the details of our response in the final version.
>
> > It would be interesting, to place the present work in perspective, to at least spend some time in related work to investigate how “explanations” are currently treated in the literature on transformer-based image models like ViTs. Is there any existing literature showing techniques for MLX (maybe not labeled as such) that are dependent on a particular architecture choice? If so, what makes the present work still interesting?
> ……..
> Attention-based architectures like ViTs, due to intermediate-layer attention maps, lend themselves to very natural investigations commensurable to the kind of pixel-wise explanations used in this work.
>
> We thank the reviewer for drawing our attention to architecture specific regularization of explanation masks. We agree that ViTs can much more readily provide a local explanation through attention maps, which were used to encode prior knowledge in the past [1, 2] (thanks for pointing us to [2]). However, we observe that attention maps are just another local explanation method like gradients or CDEP that we studied in our paper. In that spirit, we expect attention maps based explanations to have the same trend of results as the other explanation methods we studied, i.e. attention maps regularization $\leq$ perturbation-based methods $\leq$ their combination. We confirm this trend through empirical validation on the Decoy-MNIST dataset.
>
> In the shared response, we provided results when using attention maps based regularization methods proposed in [1] called SPAN and when using a Visual Transformer architecture of depth 3 and width 128. We observed that ViT architecture much more strongly latches onto the spurious correlation (when compared with Feed-Forward Network architecture that we were originally using), perhaps because of the phenomena observed in [3], making it harder to remove dependence on the decoy part. Nevertheless, we note the following trend in the results: SPAN < PGD $\leq$ PGD-Ex + SPAN. Results from our experiment are aligned with our expectations, and because attention maps are just another local explanation method, much of our intuition and takeaways continue to hold even for Visual Transformers and attention maps.
>
> More generally, we expect our method and takeaways to go stale when we have architectures that can faithfully attribute importance of different input regions (by design or otherwise) while also not compromising performance. To the best of our knowledge, no existing architecture can faithfully attribute per-region saliency. Perhaps, it is possible to come up with a carefully designed transformer architecture that can roll out per-region contributions more faithfully as the reviewer is alluding to, but such an architecture is not here yet to our knowledge. Attention maps of ViT aggregate information from other patches at each layer, which is why importance of a patch in the last layer (which is used for regularization as proposed in [1]) cannot attribute importance of a patch faithfully. Our work therefore stands relevant even with recently popular transformer-based networks .
>
> References.
> [1] Miao, Kevin, et al. "Prior Knowledge-Guided Attention in Self-Supervised Vision Transformers." arXiv preprint arXiv:2209.03745 (2022).
> [2] Yang, Ziyan, et al. "Improving Visual Grounding by Encouraging Consistent Gradient-based Explanations." Proceedings of the IEEE/CVF Conference on Computer Vision and Pattern Recognition. 2023.
> [3] Sagawa, Shiori, et al. "An investigation of why overparameterization exacerbates spurious correlations." International Conference on Machine Learning. PMLR, 2020.

---

> > ### Comment · Reviewer_WA6d · 2023-08-10
> > **Reply to the Authors**
> >
> > We thank the authors very much for their very detailed and on-point reply.
> >
> > Specifically, I found the authors to have gone above and beyond in addressing the specific point I raised on architecture-dependent MLX technique. Their new experiment compares one such technique (SPAN) with theirs, and furthermore shows that combining the two improves performance, as predicted by their previous results.
> >
> > I would thus like to raise my score to a 7 (I do not however seem to find a way to edit my own review at the moment).
> >
> > Edit: I was now able to edit the review.

---

> > > ### Author Response · Authors · 2023-08-13
> > >
> > > We are pleased to know that Reviewer WA6d found our response convincing. Thank you once again for your support and thoughtful review.

---

### Official Review · Reviewer_Hnkb · 2023-06-23

**Soundness:** 2 fair
**Presentation:** 3 good
**Contribution:** 2 fair
**Rating:** 5
**Confidence:** 4

**Summary:**

This paper reinterprets Machine learning from explanations (MLX) as a robustness problem, where human explanations define a lower-dimensional manifold for perturbations. The paper points out that the previous regularization-based MLX approaches require strong model smoothing in order to be globally effective at reducing shortcut learning. The authors propose a novel approach that combines robust training methods with an earlier MLX technique, achieving state-of-the-art results on both synthetic and real-world benchmarks. The theoretical and empirical analyses explaining how the combination of robustness and regularization can reduce the need for strong model smoothing are provided.



**Strengths:**

1. The paper is well-structured and clearly organized, making it easy to follow;

2. The authors offer theoretical analyses to explain how the combination of robustness and regularization can minimize the need for strong model smoothing, which adds to the rigor and solidity of the paper.



**Weaknesses:**

1. The contribution of the proposed method is restricted to the combination of two established robust training techniques with an existing MLX approach;

2. Although the authors thoroughly review various regularization-based MLX approaches in the introduction, the paper only showcases the effectiveness of combining robust training methods with a single MLX approach, Grad-Reg. It remains unclear whether the proposed approach can be applied to other MLX methods, and further investigation is necessary to determine its generalizability.

In addition, Table 1 does not include the results of the more recent MLX approaches that are well discussed in the introduction section.


**Questions:**

1. Is it possible to apply the proposed methods to other MLX approaches? To better understand the generalizability of the approach, it would be helpful to see the performance of robust training methods combined with various MLX methods.

2. The authors mention in lines 277-279 that they evaluate in-domain test images with background pixels replaced by a constant pixel value. Have you tested the methods' performance under other out-of-distribution (OOD) scenarios, such as replacing background pixels with the background pixels from randomly selected images?

3. A sensitivity analysis on the hyper-parameters should be provided.

**Limitations:**

See weaknesses.

At this moment, the main limitations of the paper revolve around the limited novelty and generalizability of the proposed method. I tend to reject the paper primarily for this reason. However, I may reconsider my evaluation if the authors can provide solid evidence demonstrating the generalizability of their approach.

---

> ### Author Rebuttal · Authors · 2023-08-09
>
> We thank the reviewer for their detailed review. We tried our best to address their concerns on contribution and generalizability of our method; we are more than happy to answer any further concerns.
>
> > The contribution of the proposed method is restricted to the combination of two established robust training techniques with an existing MLX approach;
>
> We wish to highlight that our contribution also lies in (a) systematic study of robustness-based methods for learning from explanations, which is missing in the existing literature, (b) theoretical analysis and experimental validation of relative merits of regularisation and robustness-based methods.
>
> Although our final recommendation of combining regularization with robustness-based methods is simple, we found consistent gains with it across multiple datasets and architectures. In that regard, we only view the simplicity of our recommendation as an appealing aspect.
>
> > It remains unclear … can be applied to other MLX methods … necessary to determine generalizability. In addition, Table 1 does not include the results of the more recent MLX approaches that are well discussed in the introduction section.
>
> We only evaluated Grad-Reg when combined with robustness methods because Grad-Reg far exceeded the performance of CDEP as reported in our results table (Table 1). “We omit comparison with Shao et al. (2021) [5] because their code is not publicly available and is non-trivial to implement the influence-function based regularization.” (L224-225). Schramowski et.al. (2020) [7] simply studied regularizing using gradient explanations (just like Grad-Reg) for supervising explanations.
>
> Nevertheless, we appreciate the concern and added in the shared response the results when combining three other MLX methods with a robustness-based method: PGD-Ex. We evaluated CDEP + PGD-Ex, Integrated-Gradient [8] and attention map using Visual Transformers (ViT) [2]. We observe from the results in the shared response that irrespective of the explanation method used, our claims remained valid: (a) using perturbations (i.e. robustness-based methods) for supervising explanations is better than using regulairzation-based methods, (b) combining robustness with regularization-based methods is at least as good or better than using robustness-based methods.
> We hope that these results with new explanation methods and other results presented in the shared response to be convincing of the generalizability of our proposal.
>
> > Have you tested under other out-of-distribution (OOD) scenarios…plant dataset
>
> We were just following the dataset construction from Shramowski et.al.[7] to replace background with the average pixel value (obtained using train split). We did not originally evaluate other shifts to tease out dependence on the background. Acting on your suggestion, we evaluated on a test set obtained by adding varying magnitude of noise to the background. We observe that robustness and regularization methods when combined led to a model that is far more robust to noise in the background, aligning with our original results on the plant dataset.
>
> |                              |  Noise (N(0, 1)) |                  | Noise (N(0, 10)) |                  | Noise (N(0, 30)) |                  |
> |------------------------------|:----------------:|:----------------:|:----------------:|:----------------:|:----------------:|:----------------:|
> |                              |      Avg Acc     |       Wg Acc     |      Avg Acc     |       Wg Acc     |      Avg Acc     |       Wg Acc     |
> | ERM                          | 59.8 ± 11.9    | 43.5 ± 2.0     | 57.4 ± 7.6     | 38.1 ± 4.8     | 55.8 ± 1.7     | 22.0 ± 3.7      |
> | Grad-Reg                     | 71.6 ± 2.0     | 66.1 ± 1.8     | **68.7 ± 6.2** | 53.4 ± 4.3     | 56.1 ± 3.3     | 34.8 ± 1.8     |
> | PGD-Ex+Grad-Reg | 69.8 ± 1.8     | **67.2 ± 2.1** | **69.5 ± 3.7** | **60.6 ± 4.8** | **67.5 ± 4.5** | **50.8 ± 2.4** |
>
> > A sensitivity analysis on the hyper-parameters should be provided.
>
> In Appendix F, Figure-4 and L668-674, we presented sensitivity analysis of PGD-Ex hyperparameters (number of steps used for optimization and epsilon) on performance for plant and ISIC dataset. Below, we also show sensitivity to hyperparameters for PGD-Ex + Grad-Reg on Decoy-MNIST dataset.
> In summary, results are broadly stable with the choice of hyperparameters, and we did not extensively search for the best hyperparameters.
>
> | Decoy-MNIST       | Lambda (Grad-Reg) | Eps (PGD-Ex) | Avg -Acc          | Wg -Acc           |
> |-------------------|-------------------|--------------|-------------------|-------------------|
> | PGD-Ex + Grad-Reg |                   |              |                   |                   |
> |                   | **1**             | **3**        | **96.9 ± 0.3** | **95.8 ± 0.4** |
> |                   | 0.1               | 3            | 96.8 ± 0.8      | 94.2 ± 0.2      |
> |                   | 5                 | 3            | 91.6 ± 0.9      | 87.6 ± 2.3      |
> |                   | 0.0001            | 3            | 75.5 ± 0.9      | 57.2 ± 3.6      |
> |                   | 1                 | 5            | 93.8 ± 1.6      | 86.3 ± 3.1      |
> |                   | 1                 | 0.1          | 58.5 ± 7.9      | 30.0 ± 2.7      |
> |                   | 1                 | 0.0001       | 59.5 ± 1.1      | 40.1 ± 2.0      |
>
> We could not share all sensitivity analysis results here due to space constraints. We will, however, make sure to include them in the final version.
>
> > … reconsider my evaluation if … provide solid evidence … generalizability
>
> We hope the reviewer finds our response to be convincing of the generalizability of our approach. We are happy to engage further to clarify any further concerns. We thank the reviewer once again for careful consideration of our paper.
>
> For references, please see our shared response.

---

> > ### Comment · Reviewer_Hnkb · 2023-08-13
> >
> > Thanks for the authors' response.
> >
> > I now well understand the two claims of the paper: (a) using perturbations (i.e. robustness-based methods) for supervising explanations is better than using regularization-based methods, (b) combining robustness with regularization-based methods is at least as good or better than using robustness-based methods alone.
> >
> > I still get some concerns and questions:
> >
> > The robustness-based methods themselves are not the contribution of this work. They are proposed in previous work, and the contribution of the paper is only to use and evaluate them in the MLX setting. Therefore, the novelty and contribution of this point appear limited in my perspective.
> >
> > Secondly, the empirical evidence presented does not consistently support the superiority of robustness-based methods. For instance, in Table 1, Grad-Reg outperforms all robustness-based methods on the Decoy-MNIST dataset. Similarly, the performance difference between robustness-based methods and Grad-Reg on the ISIC dataset is not substantial.
> >
> > Furthermore, I have reservations about the authors' implementation of CDEP. The original CDEP paper demonstrated a significant advantage of CDEP over Grad-Reg (RRR) on ISIC dataset, which is not mirrored in Table 1 of this paper. Also, according to the CDEP paper, the gap between CDEP and Grad-Reg (RRR) on the Decoy-MNIST dataset is not as large as reported in Table 1.
> >
> > Could the authors elucidate on these points? I will reassess my overall rating of the paper when more information is provided.

---

> > > ### Author Response · Authors · 2023-08-13
> > >
> > > We thank the reviewer for their response.  We are glad that our earlier response helped clear some of the concerns.
> > >
> > > > The robustness-based methods themselves are not the contribution of this work. They are proposed in this work and the contribution of the paper is only to use and evaluate them in the MLX setting. Therefore, the novelty and contribution of this point appear limited in my perspective.
> > >
> > > Yes, IBP and PGD are established, popular methods for robust-training. (One-of-)Our contribution lies in seeing their relevance in learning from explanations. Moreover, we highlight further key contributions including an analysis of regularization-based XML methods as well as proposing the novel combination of regularization-based and robustness-based methods which show consistently state-of-the-art performance.
> > >
> > > > Secondly, the empirical evidence ... not consistently support the superiority of robustness-based methods. For instance, in Table 1, Grad-Reg outperforms all robustness-based methods on the Decoy-MNIST dataset. Similarly, the performance difference between robustness-based methods and Grad-Reg on the ISIC dataset is not substantial.
> > >
> > > We agree that improvements of robustness over regularization methods are somewhat muddled by high standard deviation for Decoy-MNIST and ISIC datasets, but they were well pronounced on the plant dataset. Newly added results for Decoy-MNIST and Salient-Imagenet also substantiate the relative strength of robustness-methods over regularization-based. We also would like to again highlight consistent and considerble improvement offered by the novel combination of robustness-based and regularization-based methods.
> > >
> > > Thanks for raising this point, we will carefully rephrase the claim of robustness being better than regularization to "robustness-based methods are at least as good or better than regularization-based methods when learning from explanations".
> > >
> > > > Furthermore, I have reservations about the authors' implementation of CDEP. The original CDEP paper demonstrated a significant advantage of CDEP over Grad-Reg (RRR) on ISIC dataset, which is not mirrored in Table 1 of this paper. Also, according to the CDEP paper, the gap between CDEP and Grad-Reg (RRR) on the Decoy-MNIST dataset is not as large as reported in Table 1.
> > >
> > > *Regarding ISIC dataset discrepancy:* We thank the author for this careful observation. We readily understand the concern. We had spent significant time debugging the cause of poor performance of CDEP, our efforts and observations are documented in Appendix G: "Discussion on poor CDEP performance" of supplementary material. To summarise, we stated that the discrepancy in relative performance between Grad-Reg and CDEP (as reported by us and [1]) on the ISIC datasey may have been because they (a) use a different metric: F1, and (b) use a different architecture: VGG model pretrained on Imagenet. Furthermore, Table 4 of supplementary material shows per-group accuracy for different methods on ISIC dataset. We observe that CDEP performs well (only) on majority groups (examples without patches), which may have influenced their metrics reported in Rieger et.al. [1]
> > >
> > > *Regarding DecoyMNIST dataset discrepancy:* Decoy-MNIST setting presented in our paper is inspired from decoymnist of [1], but not the same. Sorry for any confusion. All the methods were found to be equally good on the original decoy-mnist dataset [1], which is why we had to alter the dataset to be more challenging. A key difference is that the volume of spurious/simple features in our version of decoy-mnist dataset is much higher, making it harder to remove dependence of a model on decoy/spurious features. Therefore, the performance gap reported in our paper on this dataset is not related to the one reported in CDEP paper [1], although both the datasets share the same name.
> > >
> > > *Further clarification on our implementation.* Our implementation of CDEP is borrowed from their official code repository [2]. We also made extensive search for optimal hyperparameters for CDEP, picked the best checkpoint and hyperparams using a validation set, reported avarege over three runs, and visualized its results (in Appendix G). The results reported for CDEP in the paper are our best efforts to reproduce.
> > >
> > > Thanks again.
> > >
> > > [1] Rieger, Laura, et al. "Interpretations are useful: penalizing explanations to align neural networks with prior knowledge." International conference on machine learning. PMLR, 2020.
> > > [2] deep-explanation-penalization, (2020), GitHub repository, https://github.com/laura-rieger/deep-explanation-penalization/tree/master

---

> > > > ### Comment · Reviewer_Hnkb · 2023-08-14
> > > >
> > > > I appreciate the authors' comprehensive response to my comments.
> > > >
> > > > While I maintain some reservations about the novelty of the paper, I acknowledge that most of my other concerns have been appropriately addressed.
> > > >
> > > > Consequently, I have revised my score upward to 5.

---

> > > ### Author Response · Authors · 2023-08-16
> > >
> > > We are delighted to hear that the reviewer's concerns are all resolved. We thank the reviewer for their time, passion and patience.
> > >
> > > _Regarding novelty_.
> > > We wish to highlight that robustness methods have been around since 2014 [1] and regularization of gradient explanations to learn from explanation masks was first proposed in 2017 [2]. And yet, relevance and utility of robustness methods for the learning from explanations problem has not been studied so far. Our work studied this novel combination, which filled a crucial gap and established a strong baseline for future research.
> > >
> > > We thank the reviewer once again for careful consideration of our paper.
> > >
> > > References:
> > > [1] Goodfellow, Ian J., Jonathon Shlens, and Christian Szegedy. "Explaining and harnessing adversarial examples." arXiv preprint arXiv:1412.6572 (2014).
> > > [2] Ross, Andrew Slavin, Michael C. Hughes, and Finale Doshi-Velez. "Right for the right reasons: Training differentiable models by constraining their explanations." arXiv preprint arXiv:1703.03717 (2017).

---

### Official Review · Reviewer_k7a7 · 2023-06-27

**Soundness:** 3 good
**Presentation:** 3 good
**Contribution:** 3 good
**Rating:** 7
**Confidence:** 3

**Summary:**

This paper proposes a new approach to machine learning from explanations (MLX). This new approach is still based on human-provided explanations of (ir)relevant features for each input in training, but recasts the MLX problem itself essentially into a robustness problem. The authors achieve SOTA performance when combining their method with previous ones on several benchmarks.

**Strengths:**

The authors show that the need for strong parameter smoothing of earlier approaches can be overcome, and they achieve SOTA performance on several benchmarks. Their method is intuitive and easy to understand.


**Weaknesses:**

Obtaining human-specified masks is at best a lot of effort and in many cases simply not available, which limits the scope of problems for which their method can be applied.


**Questions:**

Very minor: What is the bolding criteria for Table 1? Usually, the best model is in bold, but here there are 1, 2, or 3 models bolded depending on the dataset, and it's not clear what determined the number of bolded models for each.


**Limitations:**

It would be good to expand the "Limitations" section and provide more detail on, e.g., when the scaling breaks to large NNs, i.e. provide some guidance here about when one should expect the robustness methods to no longer be feasible to use.

---

> ### Author Rebuttal · Authors · 2023-08-09
>
> We thank the reviewer for their comments and time. We are glad that the reviewer found our method intuitive and our experiments convincing.
>
> > Obtaining human-specified masks is at best a lot of effort …
>
> We agree that manually specifying explanation masks can be impractical. However, the procedure can be automated if the nuisance/irrelevant feature occurs systematically or if it is easy to recognize, which may then be obtained automatically using a procedure similar to [1,2]. A recent effort called Salient-Imagenet used neuron activation maps to scale curation of such human-specified masks to Imagenet-scale [3, 4]. These efforts may be seen as a proof-of-concept for obtaining richer annotations beyond content labels, and towards better defined tasks.
>
> > Very minor: What is the bolding criteria for Table 1?
>
> Bold numbers in Table 1 are the ones within statistical significance bounds of the best number.
>
> > provide more detail … when the scaling breaks to large NNs…
>
> We discussed this limitation pertaining to IBP-Ex, which works by propagating axis aligned input intervals through the model. Despite its computational efficiency, IBP is known to suffer from scaling issues when the model is too big. Consequently, it is better to use IBP-Ex only when the model is small (<4 layers of CNN/feed-forward) and if computational efficiency is desired. Thanks for pointing it out, we will add this detail in the improved version of the paper.
>
> We also wish to emphasise that we do not anticipate any scaling issues when using PGD-Ex. Irrespective of the scale of the network or the size of the explanation region, we expect PGD-Ex+Grad-Reg to be at least as effective as Grad-Reg or ERM.
>
> References.
> [1] Liu, Evan Z., et al. "Just train twice: Improving group robustness without training group information." International Conference on Machine Learning. PMLR, 2021.
> [2] Rieger, Laura, et al. "Interpretations are useful: penalizing explanations to align neural networks with prior knowledge." International conference on machine learning. PMLR, 2020.
> [3] Singla, Sahil, and Soheil Feizi. "Salient ImageNet: How to discover spurious features in Deep Learning?." arXiv preprint arXiv:2110.04301 (2021).
> [4] Singla, Sahil, Mazda Moayeri, and Soheil Feizi. "Core risk minimization using salient imagenet." arXiv preprint arXiv:2203.15566 (2022).

---

> > ### Comment · Reviewer_k7a7 · 2023-08-11
> >
> > Thanks. Please consider adding most of this discussion to the paper.

---

> > > ### Author Response · Authors · 2023-08-13
> > >
> > > We are pleased to know that Reviewer k7a7 found our response satisfactory. We are grateful for your time and support.

---

### Official Review · Reviewer_haST · 2023-07-27

**Soundness:** 2 fair
**Presentation:** 3 good
**Contribution:** 2 fair
**Rating:** 5
**Confidence:** 1

**Summary:**

The paper is in the domain of MLX (machine learning from explanations). In this approach, human annotated data for each input example is available denoting features that are _relevant_ and which are _irrelevant_. It is desired that the model doesn't learn from irrelevant features.

In this paper, the authors utilize robustness for this domain. To elaborate, they expect the model to be robust to perturbations along the features which are considered irrelevant. According to the authors, this is the first use of this technique in the domain of MLX. They also combine with existing regularization-based approaches.

A theoretical framework is provided and approach evaluated on three datasets -- Decoy-MNIST, Plant, ISIC -- which are designed to capture the extent to which the models are learning from the irrelevant features. Combination of the robustness-based approaches with regularization-based approaches is shown to outperform prior approaches.

**Strengths:**

The use of robustness for MLX is well-motivated, and as per the paper, novel.

The empirical results appear to back the utility of combining robustness with existing techniques.

The paper is well-written.


**Weaknesses:**

The prior works (such as Sagawa et al., 2019; Piratla et al., 2021) have several more evaluation datasets, which the approach is not evaluated on. The generality and robustness (pardon the pun) of the approach is not completely clear.


**Questions:**

Questions
- Please address the choice of limiting the evaluation to the 3 datasets used.

Minor suggestion (no response needed)
- L63: State the domain of m^(n)
- Consider placing figure closer to their first reference. For example, Figure 2 is referenced on Pg 2 but appears on Pg 6.
- L70: "while not exploiting" -- please provide a the definition for this at this stage (addendum: if there is one at all)


**Limitations:**

Yes

---

> ### Author Rebuttal · Authors · 2023-08-09
>
> We thank the reviewer for their time and comments.
> > The prior works (such as Sagawa et al., 2019; Piratla et al., 2021) have several more evaluation datasets, which the approach is not evaluated on. … Please address the choice of limiting the evaluation to the 3 datasets used. The generality and robustness … not completely clear.
>
> Our problem setting is such that we require an input mask per training instance highlighting irrelevant features. Standard sub-population shift datasets such as the ones used in Sagawa et al., 2019; Piratla et al., 2021 do not contain any input mask, which is why we cannot evaluate them. We elaborated on differences between ours and the sub-population shift problem in L335-346 of Section 6.
>
>
> Standard datasets with input masks highlighting irrelevant regions are somewhat hard to find; to the best of our knowledge, the three datasets included were the only standard datasets that were used in the past. Nevertheless, we evaluated using a recent dataset that included relevance masks, called Salient-Imagenet. Results on Salient-Imagenet can be found in the shared response, and are in agreement with other results in the paper. We hope that the results on Salient-Imagenet and other results presented in the shared response to be convincing of the generality of our proposal.
>
> Thanks a lot for your suggestions on presentation, we will make sure to incorporate these in the final version.

---

> > ### Comment · Reviewer_haST · 2023-08-18
> >
> > Thanks a lot for your response, and the additional evaluation and insight.
> >
> > > Standard datasets with input masks highlighting irrelevant regions are somewhat hard to find
> >
> > If you have some thoughts on why this might be the case, please do share. Might it be the case that collecting such data is not easy / costly?

---

> > > ### Author Response · Authors · 2023-08-18
> > >
> > > Thanks for your response.
> > >
> > > > If you have some thoughts on why this _(standard datasets are hard to find)_ might be the case, please do share. Might it be the case that collecting such data is not easy / costly?
> > >
> > > Yes, standard datasets are hard to find partly because their curation is difficult when using a conventional annotation pipeline. However, with increasing interest in learning from explanations as a method for training reliable models [3, 4, 6, 7, 8], we are witnessing growing number of relevant datasets and techniques for efficient data curation. We answered more elaborately on advances in curation of relevant datasets in our response to Reviewer k7a7.
> > > Their question and our response to their question are relevant here, which we are pasting here for easy reference.
> > >
> > > ----
> > >
> > > Reviewer k7a7:
> > > >  Obtaining human-specified masks is at best a lot of effort …
> > >
> > > Our response:
> > > We agree that manually specifying explanation masks can be impractical. However, the procedure can be automated if the nuisance/irrelevant feature occurs systematically or if it is easy to recognize, which may then be obtained automatically using a procedure similar to [1,2]. A recent effort called Salient-Imagenet used neuron activation maps to scale curation of such human-specified masks to Imagenet-scale [3, 4]. These efforts may be seen as a proof-of-concept for obtaining richer annotations beyond content labels, and towards better defined tasks.
> > >
> > > -----
> > >
> > > We hope this answers your question. We will make sure to include these details in the main paper. We wish to also highlight that we evaluated a new dataset (Salient-Imagenet that was shared in the general response), which was inspired by your concern on standard datasets. Thanks again for your time and comment.
> > >
> > > References.
> > > [1] Liu, Evan Z., et al. "Just train twice: Improving group robustness without training group information." International Conference on Machine Learning. PMLR, 2021.
> > > [2] Rieger, Laura, et al. "Interpretations are useful: penalizing explanations to align neural networks with prior knowledge." International conference on machine learning. PMLR, 2020.
> > > [3] Singla, Sahil, and Soheil Feizi. "Salient ImageNet: How to discover spurious features in Deep Learning?." arXiv preprint arXiv:2110.04301 (2021).
> > > [4] Singla, Sahil, Mazda Moayeri, and Soheil Feizi. "Core risk minimization using salient imagenet." arXiv preprint arXiv:2203.15566 (2022).
> > > [5] Ross, Andrew Slavin, Michael C. Hughes, and Finale Doshi-Velez. "Right for the right reasons: Training differentiable models by constraining their explanations." arXiv preprint arXiv:1703.03717 (2017).
> > > [6] Pukdee, Rattana, et al. "Learning with Explanation Constraints." arXiv preprint arXiv:2303.14496 (2023).
> > > [7] Miao, Kevin, et al. "Prior Knowledge-Guided Attention in Self-Supervised Vision Transformers." arXiv preprint arXiv:2209.03745 (2022).
> > > [8] Yang, Ziyan, et al. "Improving Visual Grounding by Encouraging Consistent Gradient-based Explanations." Proceedings of the IEEE/CVF Conference on Computer Vision and Pattern Recognition. 2023.

---

> > > > ### Comment · Reviewer_haST · 2023-08-20
> > > >
> > > > Thanks, that answers my question!

---

> ### Comment · Area_Chair_hC6k · 2023-08-18
>
> Dear Reviewer haST,
>
> thanks a lot for your valuable time and comments. Since the discussion period is soon coming to an end, I wanted to ask if your concerns regarding limited evaluation have been adequately addressed by the authors or if there are any remaining weaknesses?
>
> All the best,
>
> Your AC

---

### Author Rebuttal · Authors · 2023-08-09

We thank the reviewers for detailed assessment and queries. We are glad that all the reviewers found our paper well presented and well motivated. We are also happy for an overall positive assessment of our work.

Reviewer Hnkb, WA6d, haST raised some concerns regarding overall generality of our work and raised issues regarding applicability of our proposal to alternate explanation methods (Hnkb, WA6d) or datasets (haST). We address these concerns by (a) evaluating two new explanation methods (and their combination with a robustness method: PGD-Ex) on the Decoy-MNIST dataset, (b) presenting results on a new dataset called Salient-Imagenet. We will evaluate more extensively and include these results in the final version of the paper.

In all, our response included results from two new explanation methods (Integrated-Gradient and attention map based regularization), two new architectures (Visual Transformers, ResNet-18) and one new dataset (Salient-Imagenet). As our results below demonstrate, irrespective of the explanation method or architecture or dataset, our claims remained valid: (a) using perturbations (i.e. robustness-based methods) for supervising explanations is better than using regularization-based methods, (b) combining robustness with regularization-based methods is at least as good or better than using robustness-based methods alone.

We tried our best to address all the concerns, and are more than happy to engage further to resolve any further concerns.

## Generality to new explanation methods

### Integrated-Gradient and CDEP

We introduced evaluation using Integrated-gradient [1] based regularization and also added evaluation with PGD-Ex+CDEP that we did not originally include on Decoy-MNIST dataset.

| Alg.                   | Avg Acc          | Wg Acc           |
|------------------------|------------------|------------------|
| Integrated-Grad        | 26.7 $\pm$ 1.3     | 17.6 $\pm$ 1.2     |
| CDEP                   | 14.5 $\pm$ 1.8     | 10.0 $\pm$ 0.7     |
| PGD-Ex                 | 67.6 $\pm$ 1.6     | 51.4 $\pm$ 0.3     |
| PGD-Ex+Integrated-Grad | 80.5 $\pm$ 2.1     | **62.1 $\pm$ 6.8** |
| PGD-Ex+CDEP            | **84.8 $\pm$ 0.8** | **64.2 $\pm$ 1.6** |

### Attention Map based local explanations

Using a Visual transformer architecture (of depth 3 and width 128), we evaluated regularization using local explanations obtained using an attention map – regularization based on attention maps was used to supervise prior knowledge in [2] and is called SPAN. We obtained saliency explanations on inputs using the procedure proposed in [2].

| Decoy-MNIST | Avg Acc          | Wg Acc         |
|-------------|------------------|------------------|
| ERM         | 10.0 $\pm$ 0.3     | 8.1 $\pm$ 0.3      |
| SPAN        | 19.0 $\pm$ 0.3     | 8.1 $\pm$ 0.3      |
| PGD-Ex      | **64.6 $\pm$ 4.7** | **37.4 $\pm$ 3.5** |
| PGD-Ex+SPAN | **63.1 $\pm$ 2.6** | **39.4 $\pm$ 2.9** |

## Generality to new dataset/architecture
As further evidence of generalizability of our proposal, we also evaluated on a subset of Salient-Imagenet [3, 4] using pretrained (on ImageNet) ResNet-18. Our training dataset included six classes with around 600 examples. For each example, the dataset also included a human-approved input mask highlighting spurious features. The results are as follows.

|                 | Original accuracy | Accuracy under noise |   RCS |
|-------------|------------------|------------------|-----------------------------------------------|
| ERM             |             96.43 |                87.50 | 47.88 |
| Grad-Reg        |             89.29 |                82.14 | 52.54 |
| PGD-Ex          |             93.75 |                90.18 | 58.69 |
| PGD-Ex+Grad-Reg |             94.64 |                93.75 | **65.02** |

Since the dataset did not include any natural example grouping, we could not use Worst-group accuracy metric. Instead we report using the relative core spurious (RCS) metric proposed in [4]. RCS measures relative stability of the model to noise in the core vs spurious regions. High RCS, therefore, means low dependence on spurious features. Also shown in the table are original test accuracy and accuracy when normal noise is added to spurious regions.

References.
[1] Sundararajan, Mukund, Ankur Taly, and Qiqi Yan. "Axiomatic attribution for deep networks." International conference on machine learning. PMLR, 2017.
[2] Miao, Kevin, et al. "Prior Knowledge-Guided Attention in Self-Supervised Vision Transformers." arXiv preprint arXiv:2209.03745 (2022).
[3] Singla, Sahil, and Soheil Feizi. "Salient ImageNet: How to discover spurious features in Deep Learning?." arXiv preprint arXiv:2110.04301 (2021).
[4] Singla, Sahil, Mazda Moayeri, and Soheil Feizi. "Core risk minimization using salient imagenet." arXiv preprint arXiv:2203.15566 (2022).
References.
[5] Shao, X., Skryagin, A., Stammer, W., Schramowski, P., and Kersting, K. Right for better reasons: Training differentiable models by constraining their influence functions.
[6] Stammer, W., Schramowski, P., and Kersting, K. Right for the right concept: Revising neuro-symbolic concepts by interacting with their explanations.
[7] Schramowski, P., Stammer, W., Teso, S., Brugger, A., Herbert, F., Shao, X., Luigs, H.-G., Mahlein, A.-K., and Kersting, K. Making deep neural networks right for the right scientific reasons by interacting with their explanations.
[8] Sundararajan, Mukund, Ankur Taly, and Qiqi Yan. "Axiomatic attribution for deep networks." International conference on machine learning. PMLR, 2017.

---

### Decision · Program_Chairs · 2023-09-21

**Decision:**

Accept (poster)

**Comment:**

All reviewers agree on the acceptance and the authors have been able to clarify remaining concerns. It tackles an interesting and important problem and should thus be of interest to the overall NeurIPS community